# Efficient Generative Transformer Operators for Million-Point PDEs

## Abstract

We introduce **ECHO**, a transformer–operator framework for generating million-point PDE trajectories. While existing neural operators (NOs) have shown promise for solving partial differential equations, they remain limited in practice due to poor scalability on dense grids, error accumulation during dynamic unrolling, and task-specific design. ECHO addresses these challenges through three key innovations. (i) It employs a hierarchical convolutional encode–decode architecture that achieves a 100× spatio-temporal compression while preserving fidelity on mesh points. (ii) It incorporates a training and adaptation strategy that enables high-resolution PDE solution generation from sparse input grids. (iii) It adopts a generative modeling paradigm that learns complete trajectory segments, mitigating long-horizon error drift. The training strategy decouples representation learning from downstream task supervision, allowing the model to tackle multiple tasks such as trajectory generation, forward and inverse problems, and interpolation. The generative model further supports both conditional and unconditional generation. We demonstrate state-of-the-art performance on million-point simulations across diverse PDE systems featuring complex geometries, high-frequency dynamics, and long-term horizons.

## 1 Introduction

Neural networks have emerged as a promising alternative to classical numerical solvers for modeling physical dynamics and solving partial differential equations (PDEs) (Long et al., 2018; Raissi et al., 2019; de Bézenac et al., 2019). Early data-driven approaches often targeted simple settings, with dynamics evolving on regular grids of fixed resolution. Neural operators (NOs) have overcome these constraints by learning mesh-independent functional representations that generalize across domains, resolutions, and discretizations (Lu et al., 2021; Li et al., 2021; 2023a; Serrano et al., 2023), enabling the solution of parametric (Cohen & Devore, 2015; Koupaï et al., 2024) and multi-physics problems (McCabe et al., 2024; Herde et al., 2024).

However, scaling NOs to high resolutions and large-scale realistic problems remains a fundamental challenge: interactions among all points quickly exhaust computational and memory resources. Acknowledging this limitation, current efforts to advance the field aim to generate large-scale benchmarks (Ohana et al., 2024; Elrefaie et al., 2024). Yet existing models most often consider simplified settings—either focusing on problems more amenable to efficient implementations, such as fully observed data on regular grids (Holzschuh et al., 2025; Rozet et al., 2025), or limiting the complexity of experiments when tackling cases with large-scale, high-frequency dynamics on irregular meshes. Because operating directly in the physical space rapidly becomes prohibitive, efficient neural surrogates are often based on encode–process–decode architectures operating in compressed latent spaces (Alkin et al., 2024; Serrano et al., 2024).

A first challenge is to design encoders that achieve high compression while still enabling efficient computation and preserving the relevant physical information. Current approaches leveraging spatial encoding can handle static million-point problems (Wen et al., 2025; Elrefaie et al., 2024) but fail when modeling spatio-temporal dynamics at scale. Irregular meshes commonly used in engineering applications require tracking point coordinates and neighborhoods, further increasing memory usage and accelerating saturation. Latent NOs therefore still struggle to compress spatio-temporal data effectively while preserving reconstruction quality (Koupaï et al., 2025). A second challenge lies

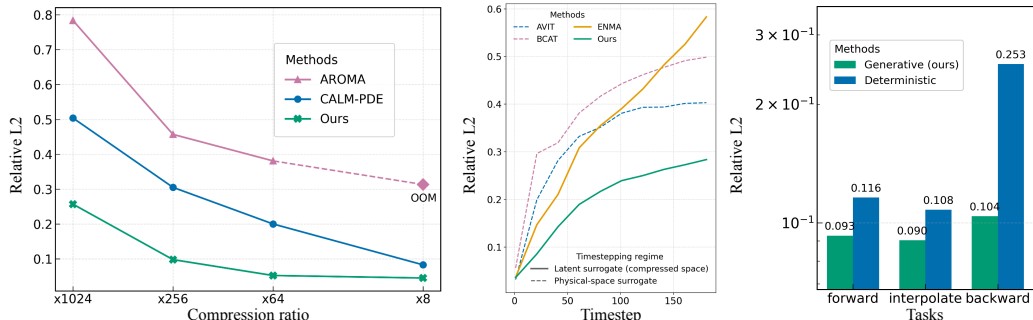

Figure 1: **Experimental analysis of ECHO.** Left: Hierarchical deep, iterative compression improves accuracy, especially at high compression ratios on *Vorticity*. Middle: ECHO's full-trajectory generation mitigates error accumulation in long-range *Gray-Scott* rollouts (40 to 160 is outside of training horizon), outperforming latent autoregressive and deterministic baselines. Right: Generative modeling consistently outperforms deterministic methods on forward, interpolation, and inverse *Rayleigh-Bénard* tasks. All plots report relative L2 error (lower is better).

in the process component, which typically relies on autoregressive time-stepping. Such rollouts accumulate errors over time, making long-horizon predictions unreliable (Lippe et al., 2023; Pedersen et al., 2025). See also the related work section in Appendix A.2. This motivates the following central question: *Can we design surrogates that achieve high compression while remaining accurate and robust on arbitrary irregular domains?* We address this through three design principles:

**(i) Hierarchical spatio-temporal compression.** Most encoders used in SOTA models downsample only the spatial dimension using e.g. heuristics neighbor aggregation (Mousavi et al., 2025; Alkin et al., 2024) or cross-attention (Serrano et al., 2024; Wang & Wang, 2024), and scale poorly with the input size. For realistic deployment, compression must act jointly on space and time. We advocate deep encoder–decoders that reduce resolution hierarchically, yielding compact yet faithful spatio-temporal latents. **(ii) Rethinking the auto-regressive process.** Next-frame(s) prediction remains the dominant training paradigm for the process (McCabe et al., 2024; Hao et al., 2024; Wang et al., 2025b; Liu et al., 2025), while suffering from error drift. We introduce a robust procedure that generates entire trajectory segments conditioned on selected frames. It captures long-range temporal dependencies and enforces horizon-wide consistency. **(iii) From deterministic to generative modeling.** We leverage a stochastic modeling formulation (Zhou et al., 2025a) for generating trajectory distributions instead of deterministic predictions that can be misleading (Huang et al., 2024; Serrano et al., 2025; Koupaï et al., 2025). This allows us to deal with partial or noisy observations, and to cope with the physical information loss inherent to the compression step.

These ideas are embodied in **ECHO**, a transformer-based operator built on an encode–generate–decode framework designed for efficient spatio-temporal PDE modeling at scale. It allows us to handle million-point trajectories on arbitrary domains. Figure 1 illustrates the benefits of principles (i)–(iii): (left) our spatio-temporal encoder achieves a compression ratio versus relative $L^2$ error that is markedly superior to state-of-the-art baselines enabling large-scale applications; (center) its trajectory-generation procedure is far less prone to error accumulation, enabling long-horizon forecasts; and (right) the generative modeling paradigm outperforms deterministic alternatives.

Additional contributions include **a staged training paradigm** that scales to dense and partially observed trajectories, overcoming memory bottlenecks faced by existing surrogates; **a unified formalism that allows solving multi-task problems**, forward and inverse problems, interpolation, and long-horizon forecasts. It enables us to address a variety of problems in a zero-shot setting.

## 2 PROBLEM SETTING

We consider time-dependent partial differential equations (PDEs) defined on a spatial domain $\Omega \subset \mathbb{R}^d$ over a time interval $[0, T]$. Each instance is specified by an initial condition $\boldsymbol{u}^0 \in L^2(\Omega, \mathbb{R}^{d_u})$ and a set of parameters $\gamma = (\boldsymbol{b}, \boldsymbol{f}, \boldsymbol{c})$, which include boundary conditions $\boldsymbol{b} \in L^2(\partial\Omega \times [0, T], \mathbb{R}^{d_b})$, a

forcing term $\boldsymbol{f} \in L^2(\Omega \times [0, T], \mathbb{R}^{d_f})$, and PDE coefficients $\boldsymbol{c}$. The governing system is

$$\mathcal{N}[\boldsymbol{u}; \boldsymbol{c}, \boldsymbol{f}](x, t) = 0, \qquad\qquad (x, t) \in \Omega \times (0, T], \qquad\qquad (1)$$

$$\mathcal{B}[\boldsymbol{u}; \boldsymbol{b}](x, t) = 0, \qquad\qquad (x, t) \in \partial\Omega \times [0, T], \qquad\qquad (2)$$

$$\boldsymbol{u}(x, 0) = \boldsymbol{u}^0(x), \qquad\qquad x \in \Omega, \qquad\qquad (3)$$

where $\mathcal{N}$ is a (possibly nonlinear) differential operator, $\mathcal{B}$ encodes the boundary conditions, and $\boldsymbol{u} : [0, T] \times \Omega \to \mathbb{R}^{d_u}$ denotes the solution field.

In contrast to classical numerical solvers, in parametric settings, fully data-driven neural solvers are not provided with an explicit PDE and must therefore be conditioned either on explicit PDE parameters $\gamma$ or on sequences of observed states. Unlike standard autoregressive approaches that advance step by step, we adopt a generative formulation that generates the full trajectory solution $\boldsymbol{u}$ at any time $t \in [0, T]$. This perspective unifies forward and inverse problems, and supports multi-task inference. For training, we assume access to a finite training set $\mathcal{D}_{\texttt{tr}}$ of $N$ trajectories, observed on a free-form spatial grid $\mathcal{X}_{\texttt{tr}}$ and discrete times $\mathcal{T} \subset [0, T]$. Each trajectory consists of $|\mathcal{X}|$ mesh points, with $|\mathcal{X}|$ potentially very large in real-world problems. At test time, trajectories are observed on a spatial grid $\mathcal{X}_{\texttt{te}}$, which may differ from $\mathcal{X}_{\texttt{tr}}$.

**ECHO** is the first generative transformer operator addressing under a unified formalism forward and inverse tasks, while operating in a compressed latent space, allowing scaling to high-resolution inputs from arbitrary domains. We first describe below the model inference setup in order to define the multi-task objectives and setting. We then introduce the architecture components (section 3.1) and the training strategy (section 3.2).

## 2.1 INFERENCE MODEL

Since directly modeling in physical space becomes computationally infeasible at large-scale, ECHO adopts an *encode–generate–decode* paradigm. Our setting (presented in fig. 2) can handle multiple situations including partial observations from regular or irregular meshes at any spatial resolution.

At inference, our objective is to generate full solution trajectories on arbitrary domains, from a limited number of observations. Let $\boldsymbol{u}^{0:T} \in \mathbb{R}^{|\mathcal{X}| \times (T+1) \times c}$ denote the full spatio-temporal trajectory, defined at $|\mathcal{X}|$ spatial locations over $T+1$ time steps with $c$ physical channels. At inference, we observe only $L$ states at arbitrary times $\{t_0, \ldots, t_{L-1}\} \subset [0, T]$. The observed subset is denoted $\mathcal{O} = \{\boldsymbol{u}^{t_\ell} : t_\ell \in \{t_0, \ldots, t_{L-1}\}\}$, and the unobserved one, $\mathcal{M} = \{\boldsymbol{u}^t : t \in [0, T] \setminus \{t_0, \ldots, t_{L-1}\}\}$, with $\mathcal{O} \cap \mathcal{M} = \emptyset$. Observations can be represented through a binary mask $m \in \{0, 1\}^{T+1}$ applied to the full trajectory, $\boldsymbol{u}^{\mathcal{O}} = \boldsymbol{u}^{0:T} \odot m$ and similarly the sequence of unobserved states can be written as $\boldsymbol{u}^{\mathcal{M}} = \boldsymbol{u}^{0:T} \odot \bar{m}$, with $\boldsymbol{u}^{\mathcal{O}} + \boldsymbol{u}^{\mathcal{M}} = \boldsymbol{u}^{0:T}$ (see fig. 2 - A).

Inference will be performed in the compressed latent space, instead of the physical space. Let $\boldsymbol{z}^{\mathcal{O}'}$ and $\boldsymbol{z}^{\mathcal{M}'}$ denote the respective latent representations of $\boldsymbol{u}^{\mathcal{O}}$ and $\boldsymbol{u}^{\mathcal{M}}$. Inference will then amount to predicting $\boldsymbol{z}^{\mathcal{M}'}$ from $\boldsymbol{z}^{\mathcal{O}'}$ and then decoding back the former to the physical space to get the reconstruction of $\boldsymbol{u}^{\mathcal{M}}$. Note that inference operates not only in a compressed spatial representation but also in a compressed temporal dimension. This process is described below.

Inputs are mapped by a hierarchical encoder $E_\phi$ onto the latent representation $\boldsymbol{z}^{\mathcal{O}'} \in \mathbb{R}^{M' \times L' \times d}$, which is defined on a regular spatial grid. $M'$ and $L'$ denote a number of spatial and temporal tokens, and $d$ is their latent dimension (see fig. 2 - B). A token here is a compressed unit representing either a spatial region or a temporal slice. The full trajectory to be generated spans the latent temporal horizon $[0, T']$, where $T'$ is a compressed representation of the physical horizon $T$, obtained through a fixed temporal downsampling factor.

The generative model $\mathcal{G}_\theta$ then predicts the complete latent trajectory $\boldsymbol{z}^{0:T'}$ via a transformer trained with a flow-matching objective, which is tractable thanks to the reduced number of tokens. During generation, unobserved latents $\boldsymbol{z}^{\mathcal{M}'}$ are initialized with Gaussian noise, while observed latents $\boldsymbol{z}^{\mathcal{O}'}$ and (if available) PDE parameters $\boldsymbol{\gamma}$ condition the process (see fig. 2 - C). Trajectories are generated by solving the ordinary differential equation (ODE):

$$\boldsymbol{z}^{0:T'} = \texttt{ODESOLVE}\Big(\mathcal{G}_\theta(\boldsymbol{z}_r^{\mathcal{M}'}, \boldsymbol{z}^{\mathcal{O}'}, \boldsymbol{\gamma}, r)\Big), \quad \boldsymbol{z}_0^{\mathcal{M}'} \sim \mathcal{N}(0, I), \qquad\qquad (4)$$

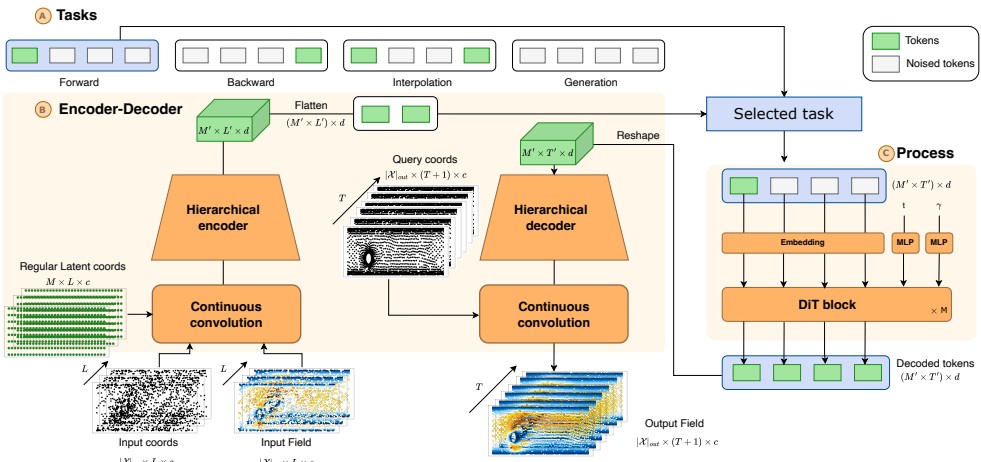

Figure 2: Architecture of the ECHO framework. ECHO comprises two components: (B) a convolutional auto-encoder and (C) a DiT-based generative process. The auto-encoder uses continuous convolutions to ingest irregular input grids of arbitrary size, map the dynamics to a regular latent grid, and hierarchically compress it; the decoder mirrors this hierarchy and applies a final continuous convolution, enabling queries at arbitrary output locations. The DiT module is trained with a flow-matching objective to denoise latent tokens, optionally conditioned on PDE parameters. This design allows ECHO to handle irregular grids and support multiple inference tasks (A).

where $r \in [0, 1]$ is the denoising index. We use the midpoint method as the ODE solver. Finally, a decoder $D_\psi$ projects the latent states back into the physical space.

ECHO supports a wide range of temporal tasks by varying the observed frames $\mathcal{O}$. For each case, inference amounts to completing a latent trajectory which is then mapped by the decoder onto the physical space. Let us consider for example a *Forward Forecasting task*. For this setting, $\mathcal{O}$ contains the first $L$ frames of a trajectory, encoded as $\boldsymbol{z}^{0:L'-1} = E_\phi(\boldsymbol{u}^{0:L-1}) \in \mathbb{R}^{M \times L' \times d}$. The model predicts the missing tokens $\boldsymbol{z}^{\mathcal{M}'} = \{\boldsymbol{z}^{L'}, \ldots, \boldsymbol{z}^{T'}\}$, completing the latent trajectory. We describe below the full generation process:

$$\boldsymbol{u}^{\mathcal{O}} \in \mathbb{R}^{|\mathcal{X}| \times L \times c} \xrightarrow{\text{encode}} \boldsymbol{z}^{\mathcal{O}'} \in \mathbb{R}^{M' \times L' \times d}$$

$$\xrightarrow{\text{mask}} \boldsymbol{z}^{\mathcal{O}' \cup \mathcal{M}'} \in \mathbb{R}^{M' \times T' \times d} \xrightarrow{\text{generate}} \hat{\boldsymbol{z}}^{O:T'} \in \mathbb{R}^{M' \times T' \times d} \xrightarrow{\text{decode}} \hat{\boldsymbol{u}}^{0:T} \in \mathbb{R}^{|\mathcal{X}| \times (T+1) \times c}$$

Additional problem instances corresponding to interpolation, inverse prediction, initial value problem and conditional / unconditional trajectory generation are described in Appendix C.1. For long-range prediction, our model naturally supports segment-wise auto-regressive generation, which we exploit to extend forecasts far beyond the training horizon. As with other neural operator approaches, our formulation operates directly on functional representations defined over arbitrary input grids, and outputs can be queried at any spatial location, enabling spatial tasks such as super-resolution and inpainting (more details in appendix C.1).

## 3 METHOD

### 3.1 ARCHITECTURE

#### 3.1.1 ENCODER–DECODER

**Encoding.** A core contribution of ECHO is its *hierarchical spatio-temporal encoder*, which progressively compresses inputs. One-shot compression, often used in encoders, degrades reconstruction quality and limits scalability. In contrast, we first embed irregular spatio-temporal data $\boldsymbol{u}^{\mathcal{O}}$ onto a fixed dense regular latent grid and then apply successive convolutional stages that reduce spatial

and temporal resolution in a structured fashion. This design achieves high compression ratios while preserving fine-scale dynamics, enabling stable and efficient generation even on million-point trajectories. More details on the encoder-decoder architecture are provided in Appendix C.2. Encoding is performed in two steps:

- **Latent grid mapping.** We first map irregular spatio-temporal inputs onto a fixed regular latent representation $\boldsymbol{u}^{\mathcal{O}}(\boldsymbol{\Xi})$, where $\boldsymbol{\Xi} = \{\xi_i\}_{i=1}^S$ is a regular grid with coordinates $\xi_i$ and $S$ is the number of grid points. This is achieved using a continuous and adaptive convolution on a real-valued function $f$ (representing any physical field variable) and a kernel $k$ (Hagnberger et al., 2025). The input field is mapped onto a latent grid composed of multiple channels:

$$(f * k)_o(\xi_j) = \sum_{i=1}^{C_{\text{in}}} \sum_{p \in \text{RF}(\xi_j)} f_i(x_p) \, k_{i,o}(\xi_j - x_p), \tag{4}$$

Here $x_p$ denotes an input position, $o$ indexes an output channel, and $\text{RF}(\xi_j)$ denotes the $P$ input points within the receptive field of the query position $\xi_j$. The kernel $k_{i,o}$ is channel-dependent and learned via a small MLP that takes as input the difference $\xi_j - x_p$ between observation point $x_p$ in the physical space and query position $\xi_j$ in the latent grid. Fixing the latent grid $\boldsymbol{\Xi}$ to be regular enables the use of standard convolutional blocks for further hierarchical compression, leading to efficient implementations. Additionally, this pointwise formulation supports chunked grid computation by processing local regions independently, thereby making calculations tractable on very dense meshes that would otherwise exceed memory limits.

- **Spatio-temporal compression.** On top of this latent representation, we employ a spatio-temporal convolutional encoder to further compress the inputs into a compact space. Directly encoding spatio-temporal dynamics is crucial for capturing motion patterns in dynamical systems. To balance efficiency and expressivity, we stack two types of blocks: `compress` blocks, which reduce either spatial or temporal dimensions, and `residual` blocks, which preserve resolution while enriching the representation. All temporal convolutions are causal, so a state at time $j$ can only attend to states at times $i \leq j$.

**Decoding** The decoder mirrors the encoder structure. Transposed convolutions reverse the spatio-temporal compression layers. The final layer is a continuous convolution layer that interpolates latent decoded tokens to arbitrary physical grids. This design allows ECHO to be queried at any location and enables the possibility to solve various spatial tasks. More details are in appendix C.2.

### 3.1.2 TRANSFORMER PROCESS

The ECHO process component is a transformer architecture trained with a flow-matching loss. It builds on the MM-DiT block (Esser et al., 2024). Given observed latent tokens $\boldsymbol{z}^{\mathcal{O}'}$, unobserved $\boldsymbol{z}^{\mathcal{M}'}$ are initialized with a noise distribution and the model is trained to reconstruct the full trajectory $\boldsymbol{z}^{0:T'}$ in the compact latent space. Thanks to the high compression ratio, global attention remains tractable—unlike recent PDE transformers that resort to neighborhood attention on raw fields (Holzschuh et al., 2025). A key feature of ECHO is its latent conditioning strategy, which is designed to support generative modeling of PDEs defined on arbitrary geometries. Instead of conditioning via adaptive normalization alone, as usually done, we concatenate the observed regularized latent tokens with noise:

$$\texttt{concat}(\boldsymbol{z}^{\mathcal{O}'}, \boldsymbol{z}^{\mathcal{M}'}_{\texttt{noise}}), \quad \text{where} \quad \mathcal{O}' \cup \mathcal{M}' = [0, T'].$$

This provides an in-context mechanism where observed frames guide noised ones. Denoising index $r$ is injected explicitly; it is embedded and added to normalization layers via AdaLN. The model thus processes the entire spatio-temporal trajectory as a unified structure. When PDE parameters $\boldsymbol{\gamma}$ are available, ECHO conditions on both $\boldsymbol{\gamma}$ and the observed tokens $\boldsymbol{z}^{\mathcal{O}'}$. Parameters are encoded by a lightweight MLP and injected into each transformer block via AdaLN (Zhou et al., 2025a; Li et al., 2025b; Zhou et al., 2025b).

### 3.2 TRAINING

Training generative models for full trajectory generation is computationally and memory intensive, particularly when handling irregular meshes, and most current methods do not scale to large size

problems. We propose an original three-stage strategy for ECHO's auto-encoder that alleviates memory constraints while preserving high-fidelity reconstruction. First, the encoder is trained to represent entire trajectory segments at low spatial resolution. Second, it is further refined to compress high-resolution individual frames. This two-step encoding approach addresses memory limitations while maintaining strong compression and accuracy. Finally, the processor is trained in this compact latent space using flow-matching. These steps are detailed below.

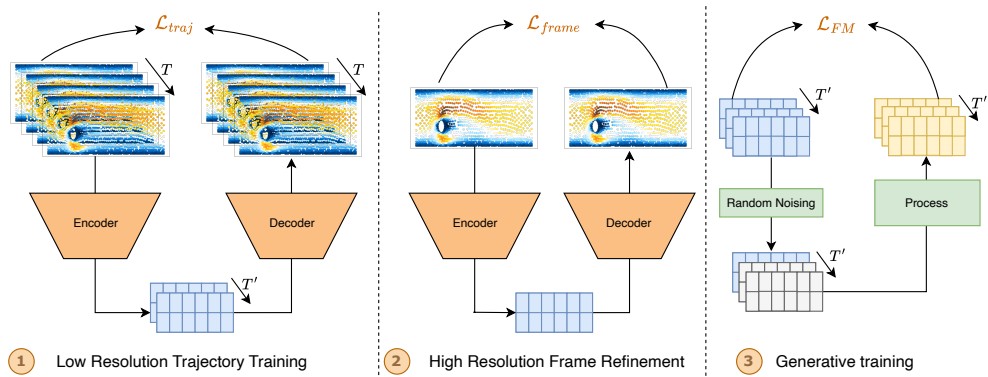

Figure 3: Three-stage training strategy for million-point trajectory generation. ECHO's auto-encoder is first trained in two steps: (1) low-resolution trajectory training and (2) high-resolution refinement on single frames. These 2 steps make use of a reconstruction objective on the input data. The generative process is then trained separately with a flow-matching objective on the encoded tokens (3), while the encoder–decoder is frozen in stage 3.

**Stage 1: Encoding - training for spatio-temporal compression on subsampled trajectories.**
Training directly on dense trajectories with many timesteps and high-resolution frames is prohibitive. To address this, we split each trajectory into sub-trajectories using a sliding window and further subsample the spatial grid at each frame by a factor between 0.2 and 0.5. This prevents out-of-memory errors while still exposing the model to relatively long temporal contexts (10–50 frames). The auto-encoder is trained with a relative mean squared error (rMSE) loss (see fig. 3 - 1):

$$\mathcal{L}_{\text{traj}} = \frac{1}{N_B} \sum_{i=1}^{N_B} \frac{\|\hat{\boldsymbol{u}}_i^{\text{trajectory}} - \boldsymbol{u}_i^{\text{trajectory}}\|}{\|\boldsymbol{u}_i^{\text{trajectory}}\|},$$

for a batch $B$ of $N_B$ trajectories.

**Stage 2: Encoding refinement - training for spatial compression on high-resolution individual frames.** The first step allows for a high spatio-temporal compression ratio but does not capture high resolution information. Since training on high resolution on whole trajectories exceeds GPU memory, we incorporate this information by training frame-wise, on high resolution frames sub-sampled between 0.5 and 1.0, instead of trajectory-wise, substantially reducing computational cost. (see fig. 3 - 2) This refinement phase is shorter than Stage 1. The loss is defined analogously on a batch $B$ of $N_B$ frames: $\mathcal{L}_{\text{frame}} = \frac{1}{N_B} \sum_{i=1}^{N_B} \frac{\|\hat{\boldsymbol{u}}_i^{\text{frame}} - \boldsymbol{u}_i^{\text{frame}}\|}{\|\boldsymbol{u}_i^{\text{frame}}\|}$.

**Stage 3: Processing - Flow-matching training.** In the final stage, the auto-encoder is frozen and the transformer processor is trained in the latent space with a flow-matching objective (detailed in section 3.1.2). To ensure robustness we proceed as follows. Consider an encoded latent trajectory $\boldsymbol{z}$, we randomly noise 80% ($\boldsymbol{z}^{\mathcal{M}'}$) of its latent frames and learn to denoise them conditioned on the observed 20% ($\boldsymbol{z}^{\mathcal{O}'}$) (see fig. 3 - 3). The goal is to learn the conditional distribution $p(\boldsymbol{z}^{\mathcal{M}'} \mid \boldsymbol{z}^{\mathcal{O}'}, \boldsymbol{\gamma})$ via a flow-based transport from a base distribution $p_0 = \mathcal{N}(0, I)$ to the target distribution. The transport is parameterized by an ODE:

$$\frac{d\mathbf{z}^r}{dr} = v(\mathbf{z}^r, r), \tag{5}$$

where $r \in [0, 1]$ is a denoising index and $v$ is a velocity field modeled by a neural network. During training, intermediate points along the probability path are sampled as:

$$\mathbf{z}^r = r\mathbf{z}^{\mathcal{M}'} + (1 - r)\boldsymbol{\epsilon}, \quad \boldsymbol{\epsilon} \sim \mathcal{N}(0, I). \tag{6}$$

The transformer, conditioned on context $\mathbf{z}^{\mathcal{O}'}$ and/or PDE parameters $\boldsymbol{\gamma}$ if available, approximates the velocity field. Its input is $(\mathbf{z}^r, \mathbf{z}^{\mathcal{O}'}, \boldsymbol{\gamma}, r)$ and its output predicts the transport direction. The flow-matching loss is:

$$\mathcal{L}_{\mathrm{FM}} = \mathbb{E}_{\boldsymbol{\epsilon},r} \left[ \left\| \mathcal{G}_\theta(\mathbf{z}^r, \mathbf{z}^{\mathcal{O}'}, \boldsymbol{\gamma}, r) - (\mathbf{z}^{\mathcal{M}'} - \boldsymbol{\epsilon}) \right\|_2^2 \right], \tag{7}$$

which encourages the model to align the predicted velocity with the true transport driving $\mathbf{z}^r$ toward $\mathbf{z}^{\mathcal{M}'}$. After training, inference proceeds as indicated in section 2.1.

## 4 EXPERIMENTS

Table 1: Datasets used in the experiments

| | Dataset | # points | grid | PDE parameters | Size ($T \times (H \times W \times D) \times C$) |
|---|---|---|---|---|---|
| 2D | Vorticity Koupaï et al. (2024) | 20M | Dense | Parametric | $20 \times \mathbf{1048576} \times 1$ |
| | Shallow-Water (Yin et al., 2022) | 2.6M | Spherical | Single instance | $40 \times 32768 \times 2$ |
| | Gray-Scott (Ohana et al., 2024) | 1.3M | - | Varying | $40 \times 16384 \times 2$ |
| | Rayleigh-Benard (Ohana et al., 2024) | 3.9M | - | Varying | $15 \times 65536 \times 4$ |
| | Acoustic Scattering Maze (Ohana et al., 2024) | 2.9M | - | Varying | $15 \times 65536 \times 3$ |
| | Active Matter (Ohana et al., 2024) | 14.4M | - | Varying | $20 \times 65536 \times \mathbf{11}$ |
| | Eagle (Janny et al., 2023) | 0.2M | Irregular | BC | $20 \times 2500 \times 4$ |
| | Cylinder Flow (Pfaff et al., 2021) | 0.3M | Irregular | Single instance | $60 \times 2000 \times 3$ |
| 3D | MHD (Ohana et al., 2024) | 37M | - | Varying | $20 \times 262144 \times \mathbf{7}$ |
| | Turbulence Gravity Cooling (TGC) (Ohana et al., 2024) | 31M | - | Varying | $20 \times 262144 \times \mathbf{6}$ |

We extensively evaluate ECHO on a diverse set of dynamical systems from public benchmarks (section 4.1), with experiments on regular grids (section 4.2) and irregular grids (section 4.3). The regular-grid benchmarks allow comparison with many state-of-the-art models that operate directly in physical space and are restricted to uniform meshes, providing evidence for our processor design (i.e. full-trajectory generation). The irregular-grid benchmarks place ECHO in a more demanding regime with complex settings (including partial observations) and varying numbers of mesh points, and are used to compare the efficiency of operator-based approaches that handle arbitrary resolutions. Finally, we further stress-test ECHO with extreme settings: long-range forecasting on turbulent dynamics beyond the training horizon (section 4.4.1), OOD evaluation on unseen physical parameters (section 4.4.2), and forward prediction on 3D datasets in section 4.5.

In appendix E, we present additional experiments: long-range prediction (appendix E.1); ECHO's generative capability and efficiency (appendix E.2); initial value problems (appendix E.3); temporal interpolation (appendix E.4); and out-of-distribution spatial inpainting (appendix E.5). We also provide ablation studies of core components of the ECHO architecture in appendix E.6.

### 4.1 DATASETS

We selected our evaluation benchmarks to highlight the flexibility of ECHO under varying conditions such as mesh resolution, time horizons, geometries, irregular grids, and parameter values. In addition, to assess ECHO's ability to handle high-resolution generation, we generated data from the vorticity equation on a dense grid of $1024 \times 1024$ points per state. Dataset characteristics are provided in table 1 and their detailed description is in appendix B.

### 4.2 EVALUATION ON REGULAR GRIDS

**Setting.** At inference, ECHO can address multiple tasks in a zero-shot manner. The comparison is performed on problems defined on regular grids, since this allows us to use SOTA models as baselines, that operate under this simplified setting which allows particularly efficient implementations. Dataset characteristics are provided on table 1. We compare against neural solvers trained for time-stepping,

the standard approach for time-dependent PDEs. The comparison is performed on forward and reverse forecasts. Since the baselines can only handle one task at a time, we train them separately for forward and backward forecasting while ECHO handles both directions without retraining. For the forward (resp. inverse) task, we use the first (resp. last) 4 timesteps as context and generate all remaining frames of the trajectory.

**Baselines.** We benchmark ECHO against both deterministic and generative baselines. Deterministic models include the transformer-based multi-physics solvers BCAT (Liu et al., 2025) and AViT (Mc-Cabe et al., 2024), Transolver++ (Luo et al., 2025), and the classical Fourier Neural Operator (FNO). We also evaluate a deterministic variant of ECHO to isolate the gains brought by the generative formulation; this variant, like ECHO, supports multi-task zero-shot inference. As a generative baseline, we consider ENMA (Koupaï et al., 2025), an autoregressive model trained with a next-token strategy akin to large language models. Implementation details for all baselines are provided in Appendix D.

Table 2: Comparison of models performance across four dynamical systems. *Determ.* = deterministic models; *Gen.* = generative models; *Inv.* = inverse task; *For.* = forward task; *Comp.* = compression ratio. All error values are Relative MSE (lower is better); "–" indicates non-convergence. Latency indicates the time for generating a whole trajectory - it is averaged here over the different datasets.

| Setting ↓ | Model ↓ | Latency (s) ↓ | Rayleigh-Benard | | | Gray-Scott | | | Active Matter | | | ASM | | |
|---|---|---|---|---|---|---|---|---|---|---|---|---|---|---|
| | | | *Inv.* | *For.* | *Comp.* | *Inv.* | *For.* | *Comp.* | *Inv.* | *For.* | *Comp.* | *Inv.* | *For.* | *Comp.* |
| Determ. | FNO | **4.42e-2** | 2.47e+3 | 4.23e-1 | ×1 | – | – | ×1 | – | – | ×1 | 1.87e0 | 1.52e0 | ×1 |
| | Trans.++ | 3.52e-1 | 6.34e-1 | 3.31e-1 | ×1 | 4.43e-1 | 2.34e-1 | ×1 | 7.33e-1 | 6.91e-1 | ×1 | 1.03e0 | 9.64e-1 | ×1 |
| | BCAT | 1.18e-1 | 1.91e-1 | 1.06e-1 | ×1 | 2.19e-1 | 8.82e-2 | ×1 | 4.98e-1 | 4.56e-1 | ×1 | 1.95e-1 | 2.18e-1 | ×1 |
| | AVIT | 1.04e-1 | 4.50e-1 | 1.01e-1 | ×1 | 1.66e-1 | 7.42e-2 | ×1 | 4.50e-1 | 4.62e-1 | ×1 | **1.04e-1** | 1.52e-1 | ×1 |
| | ECHO | 4.89e-2 | 2.53e-1 | 1.32e-1 | ×64 | 8.36e-2 | 7.66e-2 | ×32 | 5.74e-1 | 3.44e-1 | ×176 | 1.18e-1 | 2.01e-1 | ×48 |
| Gen. | ENMA | 2.20e0 | 1.71e-1 | 9.87e-2 | ×64 | 1.08e-1 | 5.44e-2 | ×32 | 4.27e-1 | 3.33e-1 | ×176 | 1.01e0 | 4.12e-1 | ×48 |
| | ECHO | **1.10e-1** | **1.16e-1** | **9.28e-2** | ×64 | **2.53e-2** | **5.12e-2** | ×32 | **1.71e-1** | **2.87e-1** | ×176 | 1.12e-1 | **1.32e-1** | ×48 |

**Results** Table 2 reports the relative MSE across four dynamical systems for both deterministic and generative surrogates. Despite operating in a highly compressed latent space - whereas all baselines except ENMA work directly in physical space without information loss - **ECHO** achieves the best overall accuracy, attaining the lowest error rates on all datasets. FNO performs poorly and fails to converge on two of the four datasets. The deterministic version of ECHO remains competitive across benchmarks, while the generative formulation consistently outperforms it, highlighting the benefits of stochastic trajectory modeling. Overall, these results underscore ECHO's ability to combine high compression with generative modeling, delivering both accuracy and scalability for million-point PDE surrogates. We also evaluate the computation time required to generate a whole trajectory (Latency column in table 2) and show that generative ECHO is on par with the deterministic baselines.

## 4.3 EVALUATION ON IRREGULAR MESHES

**Setting.** Experiments in section 4.2 show that ECHO outperforms state-of-the-art surrogates on regular grids. We now consider irregular, dense meshes, which are particularly demanding in memory and compute and require dedicated architectures. We focus on neural operators that can be queried at arbitrary resolutions and operate in a compressed latent space, comparing their encoder–decoder modules under a matched compression rate (unless stated otherwise). For each method, the encoder–decoder is first trained separately and evaluated on reconstruction to assess how well physical information is preserved. We then fix a common processor and test the expressivity of the learned latent space on a forward task (as in section 4.2). Since ECHO achieved the best scores in table 2, we use it as the processor in tables 3 and 4, so all methods generate trajectories in a one-shot manner. Evaluation is performed under two scenarios: a fully observed case, where 100% of input points are available, and a partially observed case, where only 20% of points are given and the full grid must be predicted.

**Baselines.** We benchmark ECHO against a diverse set of encoders: GINO (Li et al., 2023b), the encoder–decoder used in text2PDE (Zhou et al., 2025a); implicit neural representation models CORAL (Serrano et al., 2023); transformer-based auto-encoders AROMA and ENMA (Koupaï et al., 2025); and CALM-PDE (Hagnberger et al., 2025), a continuous-convolution model. In table 3, we contrast hierarchical compression (ENMA, CALM-PDE, ECHO) with direct latent compression

(GINO, CORAL, AROMA). All methods use a similar compression ratio to ensure fairness, except if it could not fit memory constraints. Training details are provided in Appendix D.

Table 3: Comparison of encoder–decoder modules from operator surrogates on irregular scenarios. Metrics are reported as Relative MSE on the test set; *Rec.* and *For.* denote reconstruction and forward generation. Best and second-best scores are in **bold** and underlined. Colors indicate encoder scalability: green = full-trajectory training without issues; yellow = architectural tweaks required to encode full trajectories (see appendix D); red = smaller tokens or models needed to fit a full trajectory in GPU memory. Column *Irr.* specifies the strategy for handling irregular points. Column $\mathcal{X}_{te}\downarrow$ gives the spatial sampling ratio of conditioning frames at inference (100% uses the full training resolution; 20% uses a random 20% of grid points). All metrics are computed on the full grid.

| $\mathcal{X}_{te}\downarrow$ | dataset | $\rightarrow$ | Vorticity | | Shallow-Water | | Eagle | | Cylinder Flow | |
|---|---|---|---|---|---|---|---|---|---|---|
| | Irr. | Encoder | *Rec.* | *For.* | *Rec.* | *For.* | *Rec.* | *For.* | *Rec.* | *For.* |
| 100% | Graph | GINO | 9.99e-1 | 1.00 | 8.68e-1 | 1.11 | 5.58e-1 | 1.89 | 7.94e-1 | 8.65e-1 |
| | INR | CORAL | 5.13e-1 | 1.34 | 2.29e-1 | 6.89e-1 | 6.04e-1 | 1.54 | 2.94e-1 | 5.08e-1 |
| | Attention | AROMA | 5.13e-1 | 8.42e-1 | 2.51e-2 | 4.21e-2 | 2.81e-1 | 3.09e-1 | **2.76e-2** | 2.21e-1 |
| | | ENMA | 4.38e-1 | 4.38e-1 | 7.17e-2 | 7.35e-2 | 2.85e-1 | 3.29e-1 | 7.50e-2 | 1.04e-1 |
| | Convolution | CALM-PDE | 2.69e-1 | 4.20e-1 | 2.56e-1 | 2.80e-1 | 9.01e-1 | 9.24e-1 | 2.44e-1 | 3.56e-1 |
| | | ECHO | **6.88e-2** | **1.71e-1** | **1.78e-2** | **1.96e-2** | **1.71e-1** | **2.55e-1** | 4.68e-2 | **8.82e-2** |
| 20% | Graph | GINO | 9.99e-1 | 1.00 | 8.73e-1 | 1.71 | 6.77e-1 | 1.71 | 7.95e-1 | 8.65e-1 |
| | INR | CORAL | 5.19e-1 | 1.33 | 2.29e-1 | 6.87e-1 | 6.31e-1 | 1.54 | 3.31e-1 | 5.18e-1 |
| | Attention | AROMA | 5.14e-1 | 8.43e-1 | 3.40e-2 | 4.57e-2 | 4.01e-1 | 4.83e-1 | **4.59e-2** | 2.27e-1 |
| | | ENMA | 4.41e-1 | 4.41e-1 | 8.16e-2 | 8.09e-1 | 3.66e-1 | **3.99e-1** | 8.68e-2 | 1.20e-1 |
| | Convolution | CALM-PDE | **2.95e-1** | 4.38e-1 | 2.77e-1 | 2.95e-1 | 9.61e-1 | 9.75e-1 | 2.68e-1 | 3.77e-1 |
| | | ECHO | 3.66e-1 | **2.21e-1** | **2.61e-2** | **2.61e-2** | 2.46e-1 | 4.72e-1 | 5.51e-2 | **1.03e-1** |

**Results** These experiments highlight ECHO's strong and consistent performance across all settings. In contrast, graph-based and INR approaches fail to encode the complex dynamics considered here. Attention-based methods, such as AROMA, are powerful encoders—ranking second in most experiments and excelling on smaller datasets like *Cylinder Flow* - but their high memory requirements prevent scaling to larger datasets, as shown by their weaker performance on *Vorticity*. Conversely, the convolution-based CALM-PDE performs better on *Vorticity* due to its lighter memory footprint. Together with ECHO, these findings demonstrate that hierarchical convolutional architectures enhance performance, particularly on large-scale datasets. Operating in a compressed latent space—both spatially and temporally—reduces error accumulation during generation. This advantage is most evident on *Cylinder Flow*, where AROMA achieves the best reconstruction performance, but ECHO surpasses it in the forward task accuracy.

Table 4: Super-resolution (forecast of 200% of the grid) Forward generation (For.) on a $1024 \times 1024$ **Vorticity** grid. Metric: Relative MSE (lower is better). All encoder baselines are out-of-memory (OOM) except ECHO.

| Task | GINO | CORAL | AROMA | CALM-PDE | Transolver++ | ECHO |
|---|---|---|---|---|---|---|
| *For.* | OOM | OOM | OOM | OOM | OOM | **3.88e-1** |

**Results** Table 4 presents results for a spatio-temporal forecasting task on our large-scale *Vorticity* dataset with a grid size of $1024 \times 1024$. All baselines ran out of memory on the H100 GPU with 80 GB used for these experiments; only ECHO scaled successfully.

## 4.4 OUT-OF-DISTRIBUTION EVALUATION

### 4.4.1 LONG-RANGE PREDICTION

We further illustrate ECHO's ability to generate long-range predictions beyond the training horizon on the Vorticity dataset, which exhibits turbulent regimes. Figure 4a and table 4b illustrate that ECHO remains consistent in long-rollouts, even on turbulent dynamics such as the Vorticity PDE. Note that we observed in fig. 1 (middle) that ECHO accumulates less error than baselines.

(b) Horizon Analysis outside of the training horizon. $T$ is the training horizon. Metric is the Relative MSE

| Horizon | $0.5T$ | $T$ | $1.5T$ | $2T$ | $2.5T$ |
|---------|--------|-----|--------|------|--------|
| **MSE** | 7.7e-2 | 1.72e-1 | 2.62e-1 | 3.79e-1 | 4.79e-1 |

(c) OOD Generalization on unseen viscosities. Metric is the Relative MSE.

| Setting | In-dist. | OOD 0-shot | OOD few-shot |
|---------|----------|------------|--------------|
| **MSE** ↓ | 1.17e-1 | 3.39e-1 | 1.12e-1 |

(a) Qualitative visualization of long-range generation on the Vorticity equation. The model is trained on trajectories of length 20 time steps and evaluated beyond this horizon.

Figure 4: Vorticity analysis: (a) qualitative ground-truth vs ECHO long-range rollouts beyond the training horizon; (b) relative MSE across prediction horizons; (c) in-distribution and OOD generalization.

### 4.4.2 NEW PHYSICAL PARAMETERS

In this section, we probe ECHO's ability to (i) generalize to out-of-distribution PDE parameters and (ii) specialize at a specific task (Forward generation here) on new data. We vary the viscosity parameter: training uses $\nu \in [10^{-3}, 10^{-4}]$, while OOD evaluation uses $\nu \in [10^{-4}, 10^{-5}]$, yielding more turbulent, higher-Reynolds flows with different initial conditions. We consider two settings: (i) a strict **zero-shot** scenario, where the pre-trained model is directly applied to the new regime, and (ii) a **few-shot adaptation** scenario, where the model is lightly fine-tuned on a small set of OOD trajectories for the forward prediction task.

**Results** As shown in Table 4c, ECHO displays strong zero-shot transfer to the more turbulent regime. Modest few-shot fine-tuning on OOD forward trajectories further improves accuracy, illustrating how a multi-task pre-trained operator can be efficiently adapted to new dynamics, paving the way toward universal PDE surrogates that are trained once and quickly specialized at low cost.

### 4.5 EVALUATION ON 3D REGULAR DYNAMICAL SYSTEMS

To further assess the scalability of ECHO, we test it on 3D spatio-temporal systems. The datasets considered are described in table 5 and appendix B.

**Results** ECHO successfully scales to 3D scenarios compared to existing baselines. Its full-trajectory generation design accumulates less errors than other baselines. In particular, we observe that FNO performs very well on first frame generation, but quickly diverges as the dynamic evolve, whereas ECHO provides consistent prediction along the entire trajectory.

Table 5: Forward time-stepping on 3D Benchmarks. Relative MSE across horizons.

| Model ↓ | MHD | | | TGC | | |
|---------|------|------|------|------|------|------|
| | *1st* | *0.5T* | *T* | *1st* | *0.5T* | *T* |
| FNO | **1.69e-1** | – | – | **3.15e-2** | – | – |
| AVIT | 3.40e-1 | 5.68e-1 | 7.51e-1 | 1.26e-1 | 6.22e-1 | 1.32e0 |
| BCAT | 3.29e-1 | 5.43e-1 | 6.70e-1 | 1.24e-1 | 9.89e-1 | 1.62e0 |
| ECHO | 2.70e-1 | **3.82e-1** | **5.50e-1** | 1.13e-1 | **5.55e-1** | **6.67e-1** |

## 5 CONCLUSION

We introduced ECHO, a transformer–operator framework for large-scale, high-resolution physical simulations. ECHO builds on three core ideas: (i) high compression via a hierarchical spatio-temporal encoder that efficiently handles million-point datasets; (ii) a generative modeling paradigm that learns entire trajectory segments, mitigating long-horizon error drift; and (iii) a unified training strategy that decouples representation learning from task supervision. Operating in a highly compressed latent space, ECHO effectively models complex PDE systems with intricate geometries and high-frequency dynamics on irregular meshes. It extends the trade-off between compression ratio and predictive quality, enabling truly large-scale applications. Experiments show that ECHO achieves a superior compression–accuracy balance, substantially reduces error accumulation compared to autoregressive baselines, and outperforms deterministic approaches across forward prediction, interpolation, and inverse problem-solving tasks.

ETHICS STATEMENT

PDEs are involved in many applications of science and engineering. This paper introduces a new method to improve the performance and scalability of surrogate models for solving PDEs. While we do not directly target such real-world applications in this paper, one should acknowledge that solvers can be used in a wide range of scenarios including weather, climate, medical, aerodynamics, industry, and military applications. We used ChatGPT to polish the writing.

REPRODUCIBILITY STATEMENT

We provide implementation details in appendix D including architecture details, training hyper-parameters and baseline descriptions. Moreover, we will publicly release the code base and the generated dataset upon acceptance. All our experiments run on Nvidia H100 GPUs with 80Gb memory.

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

APPENDIX: TABLE OF CONTENTS

# A  RELATED WORKS

Table 6: Comparison of operator learning approaches for spatiotemporal PDE forecasting.

| Model | | Reference | 1. Arbitrary Domains | 2. Supports Compression | 3. High-Resolution Encoding | 4. Hierarchical Encoding | 5. Generative modeling | 6. Multi-task prediction |
|---|---|---|---|---|---|---|---|---|
| Graph | MPPDE | Brandstetter et al. (2022) | ✓ | ✗ | ✗ | ✗ | ✗ | ✗ |
| | RIGNO | Mousavi et al. (2025) | ✓ | ✓ | ✗ | ✗ | ✗ | ✗ |
| Operator | Transolver++ | Luo et al. (2025) | ✓ | ✗ | ✓ | ✗ | ✗ | ✗ |
| | GINO | Li et al. (2023b) | ✓ | ✓ | ✗ | ✗ | ✗ | ✗ |
| | TEXT2PDE | Zhou et al. (2025a) | ✓ | ✓ | ✗ | ✓ | ✓ | ✗ |
| INRs | CORAL | Serrano et al. (2023) | ✓ | ✓ | ✗ | ✗ | ✗ | ✗ |
| | DiNo | Yin et al. (2022) | ✓ | ✓ | ✗ | ✗ | ✗ | ✗ |
| Attention | UPT | Alkin et al. (2024) | ✓ | ✓ | ✗ | ✗ | ✗ | ✗ |
| | ENMA | Koupaï et al. (2025) | ✓ | ✓ | ✗ | ✓ | ✓ | ✗ |
| Convolution | CALM-PDE | Hagnberger et al. (2025) | ✓ | ✓ | ✗ | ✓ | ✗ | ✗ |
| | ECHO | Ours | ✓ | ✓ | ✓ | ✓ | ✓ | ✓ |

## A.1  OPERATOR LEARNING

Operator learning has emerged as a powerful paradigm for modeling mappings between infinite-dimensional function spaces, enabling neural surrogates to learn the solution operators of partial differential equations (PDEs) directly. Early breakthroughs such as DeepONet and the Fourier Neural Operator (FNO) established neural operators (NOs) as effective and mesh-independent tools for approximating PDE solution maps (Li et al., 2021; Lu et al., 2021). Since then, research has focused on improving both the expressiveness and scalability of NOs. For example, multi-scale modeling has been explored through factorized representations and wavelet-based decompositions (Gupta et al., 2021), while latent-space formulations have been introduced to support complex and irregular geometries (Tran et al., 2023; Li et al., 2023b). Implicit neural representations (INRs), as in CORAL, have also been proposed to flexibly handle variable spatial discretizations at inference time (Yin et al., 2022; Serrano et al., 2023; Wang et al., 2025a).

A major trend in recent years has been the adoption of *attention-based architectures* for operator learning on irregular meshes. OFormer introduced transformers for embedding input–output function pairs, demonstrating their strong representation capabilities for operator approximation (Li et al., 2023a). This line of work inspired more advanced designs such as GNOT and Transolver, which refine input encoding and generalization to complex domains (Hao et al., 2023; Wu et al., 2024). However, transformer-based NOs often suffer from quadratic complexity and memory usage, making them difficult to scale to real-world scenarios where spatial resolutions can reach millions of points.

These limitations have motivated the development of *efficient neural operator surrogates* aimed at reducing computational and memory costs without sacrificing accuracy. Perceiver-inspired approaches such as Aroma and UPT learn compact latent representations that decouple the number of tokens from the raw input size, significantly improving scalability (Serrano et al., 2024; Alkin et al., 2024). While these methods achieve strong results on moderately large inputs, our empirical analysis in fig. 1 (left) shows that attention-only latent encoders degrade at extreme compression ratios, limiting their effectiveness for high-resolution generative modeling.

A complementary direction has explored *hierarchical compression*. CALM-PDE (Hagnberger et al., 2025) demonstrated that progressively reducing spatial resolution via continuous convolutions improves efficiency while preserving fidelity. Building on this idea, we introduce a spatio-temporal hierarchical encoder that extends this principle beyond static fields to dense time-dependent trajectories. By gradually compressing both space and time from a dense latent grid to coarser levels, our approach constructs highly informative and compact latent spaces, enabling efficient generative modeling even on million-point spatio-temporal inputs.

## A.2  GENERATIVE MODELS

While most operator learning methods are deterministic, generative modeling introduces key capabilities for physical systems—most notably the ability to capture uncertainty and represent one-to-many mappings. This is especially valuable in chaotic or partially observed regimes, where determinis-

tic predictions can quickly diverge. Two main generative paradigms have emerged for scientific modeling: diffusion-based methods and autoregressive transformers.

**Diffusion Transformers.**   Diffusion models synthesize data by learning to reverse a progressive noising process through a sequence of denoising steps (Ho et al., 2020). In computer vision, Latent Diffusion Transformers (DiTs) extend this approach by operating in the latent space of a variational autoencoder (VAE), achieving strong generative quality and scalability (Peebles & Xie, 2022).

Recently, diffusion has been adapted to physical modeling. Kohl et al. (2024) introduced an autoregressive diffusion framework for PDE dynamics, aiming to better capture stochasticity in turbulent flows. Lippe et al. (2023) improved the treatment of chaotic high-frequency regimes by modulating noise variance during denoising. Generative diffusion is particularly attractive under partial observations or incomplete physics (Huang et al., 2024). For example, Li et al. (2025a) reformulate PDE solving as a video inpainting problem to handle both forward and inverse tasks, but the method remains limited to regular grids and is expensive at inference. Zhou et al. (2025a) further extend DiTs to physics-driven data generation from textual prompts.

Despite their flexibility, diffusion models are computationally heavy at inference due to the large number of denoising steps. *Flow Matching* provides a more efficient alternative by learning a continuous-time velocity field that transports one distribution to another via an ordinary differential equation (ODE), enabling much faster sampling with far fewer steps (Lipman et al., 2023; 2024). Flow matching has recently been applied to PDE surrogates (Holzschuh et al., 2025), mostly in multi-physics settings and under fixed, regular discretizations. More recently, (Li et al., 2025b), proposed to use Flow matching to solve PDE framewise. The main difference with our proposed method lies in the formulation that we adopted, that allows us to solve several tasks without retraining. Moreover, ECHO makes use of continuous convolution, allowing our model to solve PDE on very dense grids.

**Autoregressive Transformers.**   Autoregressive (AR) models, originally designed for language modeling, have been extended to visual domains by sequentially generating spatial and temporal tokens. A common design couples a vector-quantized variational autoencoder (VQ-VAE) with a causal or bidirectional transformer to model discrete token sequences (van den Oord et al., 2017; Esser et al., 2021; Chang et al., 2022). In video generation, *Magvit* and *Magvit2* (Yu et al., 2023; 2024) use 3D CNNs to encode spatiotemporal structure and autoregressively predict quantized latent tokens for future frames.

This paradigm has recently been adapted to PDE surrogates. *Zebra* (Serrano et al., 2025) combines a spatial VQ-VAE with a causal transformer for in-context forecasting of dynamical systems. However, reliance on discrete codebooks can limit expressiveness and hinder accurate representation of fine-grained continuous physical phenomena. Nguyen et al. (2025) extend this design to multi-physics settings by adding a decoder refinement stage to reduce quantization artifacts, at the cost of extra complexity. Alternatively, Koupaï et al. (2025) replace discrete tokens with continuous ones and train next-token generation via a flow-matching objective.

Yet, whether discrete or continuous, autoregressive strategies remain prone to long-horizon error accumulation, particularly in compressed latent spaces where information loss amplifies drift from the true solution (Pedersen et al., 2025). Our work departs from the next-step paradigm: we train a flow-matching transformer to synthesize entire trajectory segments directly rather than predicting one token at a time. This design mitigates temporal drift and enables more stable long-range rollouts than both diffusion-based and deterministic baselines (see the right panel of fig. 1).

# B DATASET DETAILS

## B.1 VORTICITY

We consider a 2D turbulence model and focus on the evolution of the vorticity field $\omega$, which captures the local rotation of the fluid and is defined as $\omega = \nabla \times \mathbf{u}$, where $\mathbf{u}$ is the velocity field. The governing equation is:

$$\frac{\partial \omega}{\partial t} + (\mathbf{u} \cdot \nabla)\omega - \nu \nabla^2 \omega = 0, \tag{8}$$

where $\nu$ denotes the kinematic viscosity, defined as $\nu = 1/\mathrm{Re}$. We sample $\nu \sim \mathcal{U}([10^{-3}, 10^{-2}])$. The initial conditions are generated from the energy spectrum:

$$E(k) = \frac{4}{3}\sqrt{\pi}\left(\frac{k}{k_0}\right)^4 \frac{1}{k_0} \exp\left(-\left(\frac{k}{k_0}\right)^2\right), \tag{9}$$

where $k_0$ denotes the characteristic wavenumber. Vorticity is linked to energy by the following equation :

$$\omega(k) = \sqrt{\frac{E(k)}{\pi k}} \tag{10}$$

We generate very dense-grid with $1,024 \times 1,024$ spatial resolution and 20 timesteps, resulting in trajectories with more than 20M points.

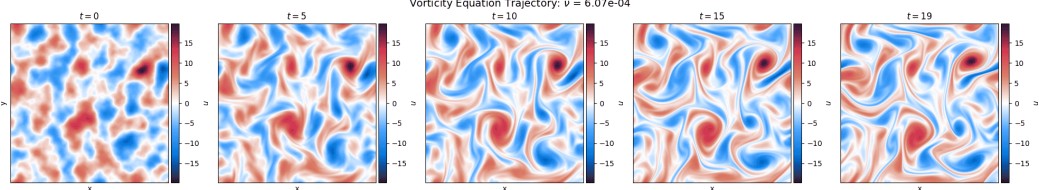

Figure 5: Sample of the Vorticity dataset.

## B.2 SHALLOW-WATER

**3D-Spherical Shallow-Water** (*Shallow-Water*). We consider the shallow-water equations on a rotating sphere, modeling large-scale atmospheric flows:

$$\frac{\partial u}{\partial t} = -f \cdot k \times u - g\nabla h + \nu \Delta u, \tag{11}$$

$$\frac{\partial h}{\partial t} = -h\nabla \cdot u + \nu \Delta h, \tag{12}$$

where $k$ is the outward unit normal to the sphere, $u$ is the tangential velocity field (with vorticity $w = \nabla \times u$), and $h$ denotes the fluid height.

Data is generated with the *Dedalus* software (Burns et al., 2020), following Yin et al. (2022). The setup produces symmetric dynamics in both hemispheres. The initial zonal velocity $u_0$ contains two symmetric jets parallel to latitude circles:

$$u_0(\phi, \theta) = \begin{cases} \left(\frac{u_{max}}{e_n} \exp\left(\frac{1}{(\phi-\phi_0)(\phi-\phi_1)}\right), 0\right), & \phi \in (\phi_0, \phi_1), \\ \left(\frac{u_{max}}{e_n} \exp\left(\frac{1}{(\phi+\phi_0)(\phi+\phi_1)}\right), 0\right), & \phi \in (-\phi_1, -\phi_0), \\ (0, 0), & \text{otherwise}, \end{cases} \tag{13}$$

where $\phi$ and $\theta$ are latitude and longitude, $u_{max}$ is the maximum velocity, $\phi_0 = \frac{\pi}{7}$, $\phi_1 = \frac{\pi}{2} - \phi_0$, and $e_n = \exp\left(-\frac{4}{(\phi_1-\phi_0)^2}\right)$.

The initial fluid height $h_0$ is computed by solving a boundary value problem as in Galewsky et al. (2004), perturbed with:

$$h_0'(\phi, \theta) = \hat{h} \cos(\phi) \exp\left(-\left(\tfrac{\theta}{\alpha}\right)^2\right) \left[\exp\left(-\left(\tfrac{\phi_2-\phi}{\beta}\right)^2\right) + \exp\left(-\left(\tfrac{\phi_2+\phi}{\beta}\right)^2\right)\right], \qquad (14)$$

with constants $\phi_2 = \tfrac{\pi}{4}$, $\hat{h} = 120$ m, $\alpha = \tfrac{1}{3}$, and $\beta = \tfrac{1}{15}$ (Galewsky et al., 2004). Simulations are run on a latitude–longitude grid of size $128 \times 256$.

For data generation, $u_{max}$ is sampled from $\mathcal{U}(60, 80)$. Trajectories span 320 hours with hourly snapshots, yielding 320 frames each. To emphasize meaningful dynamics, we retain only the last 160 frames. Each long trajectory is split into sub-trajectories of 40 frames. The resulting dataset comprises 64 training trajectories and 8 test trajectories. Finally, we rescale the fields for numerical stability: heights $h$ by $3 \times 10^3$ and vorticity $w$ by 2.

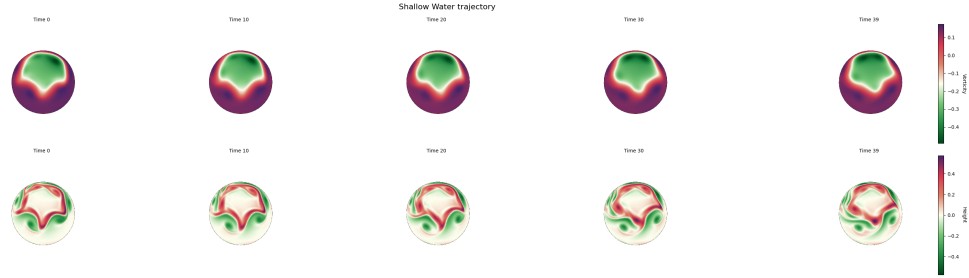

Figure 6: Sample of the Shallow-water dataset.

### B.3 EAGLE

Eagle is taken from (Janny et al., 2023) and represents the fluid velocity and pressure around a moving source. This dataset has different boundary shapes that create varying flow around the source. Moreover, trajectories are long (990 timesteps), making this dataset well-suited for long-rollout tasks. For additional deatails on dataset generation, we refer readers to (Janny et al., 2023). We provide a visualization of a trajectory in fig. 7

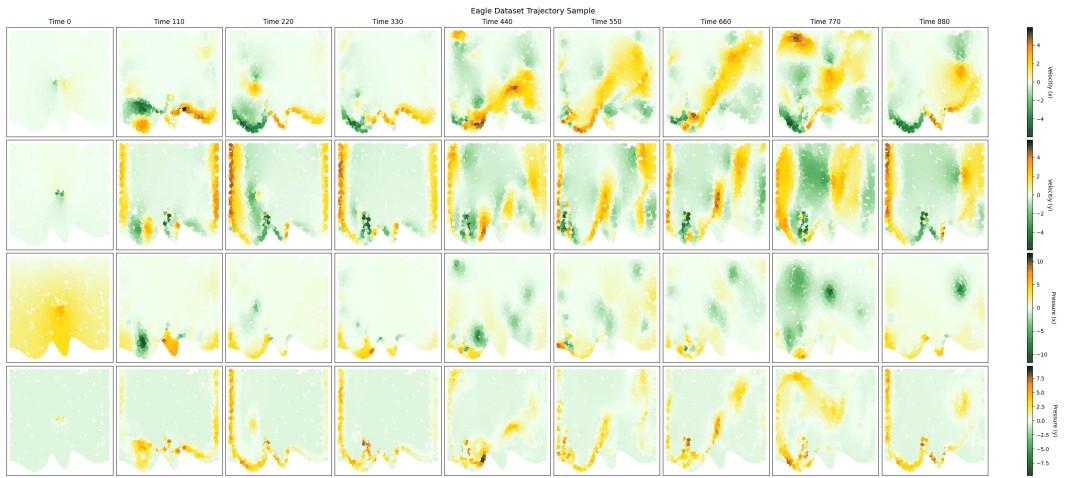

Figure 7: Sample of the Eagle dataset.

### B.4 CYLINDER-FLOW

We use the dataset introduced by Pfaff et al. (2021), which simulates water flow in a channel with a cylinder serving as an obstacle. The governing equations are the 2D incompressible Navier–Stokes

equations with constant density:

$$\partial_t \mathbf{v} = 0, \tag{15}$$

$$\rho_0 \left( \partial_t \mathbf{v} + \mathbf{v} \cdot \nabla \mathbf{v} \right) + \nabla p = \mu \nabla^2 \mathbf{v}, \tag{16}$$

$$\mathbf{v} := \mathbf{v}(t, \omega), \quad p := p(t, \omega), \quad \omega \in \Omega, \, t \in [0, T], \tag{17}$$

where $\rho_0$ denotes the constant density, $\mathbf{v}$ the velocity field, and $p$ the pressure. Each sample varies in the cylinder's diameter and position, and the task of neural surrogates is to predict both velocity and pressure fields. For our experiments, we subsample trajectories to 15 timesteps. Further details about the setup can be found in Pfaff et al. (2021).

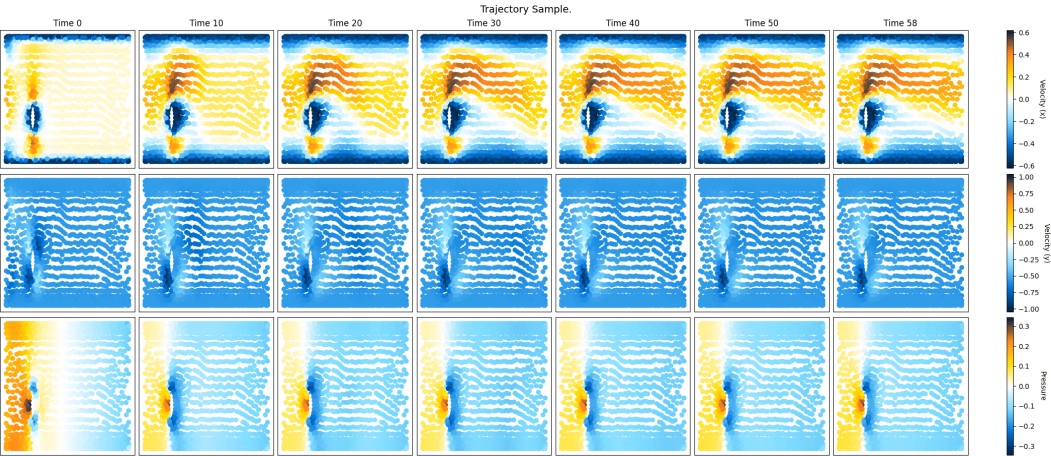

Figure 8: Sample of the Cylinder Flow dataset.

### B.5 RAYLEIGH-BÉNARD

We consider a 2D horizontally periodic fluid subject to buoyancy-driven convection. The dataset is taken from (Ohana et al., 2024), which provides the full setting for data generation. The state variables are the velocity $u = (u_x, u_z)$, buoyancy $b$, and pressure $p$. Heating from below and cooling from above create density variations that drive convection, producing characteristic Bénard cells where hot fluid rises and cold fluid sinks.

The dynamics are governed by

$$\frac{\partial b}{\partial t} - \kappa \Delta b = -u \cdot \nabla b, \tag{18}$$

$$\frac{\partial u}{\partial t} - \nu \Delta u + \nabla p - b e_z = -u \cdot \nabla u, \tag{19}$$

with $\Delta = \nabla \cdot \nabla$ the Laplacian, $e_z$ the vertical unit vector, and $\int p = 0$ a gauge constraint. The first equation describes convection–diffusion of buoyancy, while the second is the Navier–Stokes equation with buoyancy forcing. The domain is periodic in $x$ with vertical boundary conditions $u(z = 0) = u(z = L_z) = 0$ and $b(z = 0) = b(z = L_z) = 0$. The system is parameterized by the Rayleigh and Prandtl numbers through thermal diffusivity $\kappa$ and viscosity $\nu$:

$$\kappa = (\text{Rayleigh} \times \text{Prandtl})^{-1/2}, \quad \nu = \left( \frac{\text{Rayleigh}}{\text{Prandtl}} \right)^{-1/2}. \tag{20}$$

Here, the Rayleigh number measures the ratio of buoyancy to diffusion forces, while the Prandtl number controls the balance between momentum and thermal diffusion. We provide in fig. 9 visualizations of 2 trajectories.

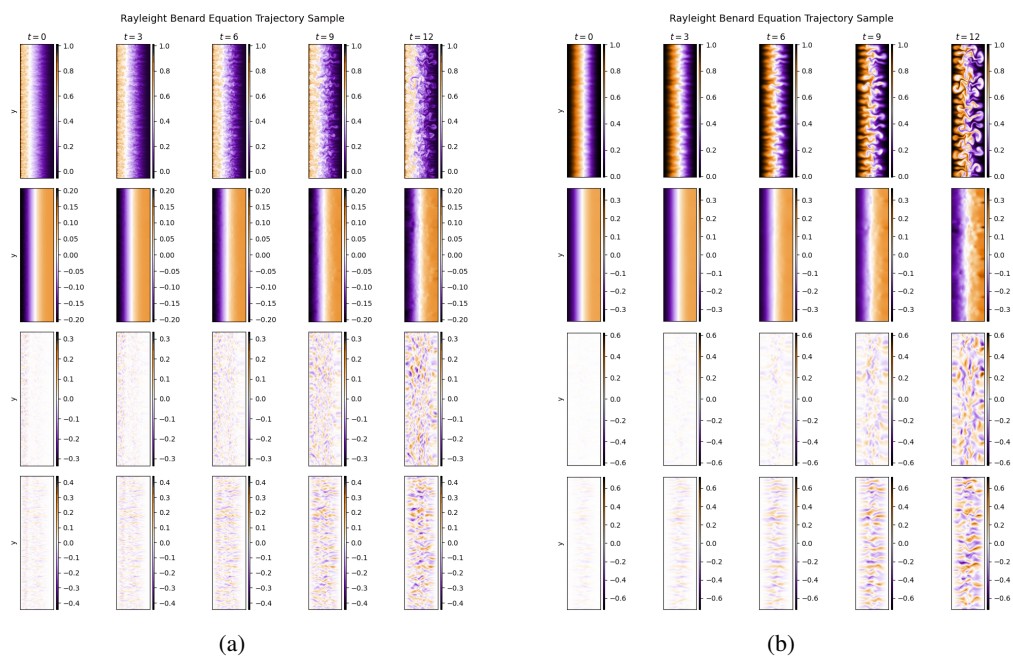

(a)                                                 (b)

Figure 9: Samples from the *Rayleigh-Bénard* dataset.

## B.6 GRAY-SCOTT

The Gray–Scott system (Pearson, 1993) is a pair of coupled reaction–diffusion equations modeling the interaction of two chemical species, $A$ and $B$, whose concentrations evolve in space and time:

$$\frac{\partial A}{\partial t} = \delta_A \Delta A - AB^2 + f(1-A), \tag{21}$$

$$\frac{\partial B}{\partial t} = \delta_B \Delta B + AB^2 - (f+k)B. \tag{22}$$

Here, $f$ and $k$ are the *feed* and *kill* rates: $f$ controls the supply of species $A$, while $k$ regulates the removal of species $B$. The diffusion coefficients $\delta_A$ and $\delta_B$ determine the transport rates of the two species.

The dataset used in our experiments is provided by Ohana et al. (2024), which includes further details on its generation process. See fig. 10 for a visualization of a trajectory.

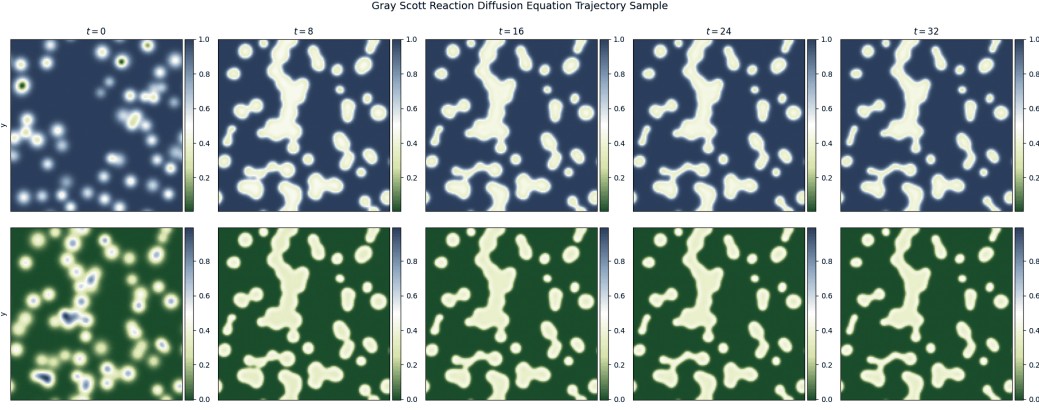

Figure 10: Sample of the *Gray-Scott Reaction-Diffusion* dataset

## B.7 ACTIVE MATTER

Active Matter features simulations of rod-like biological active particles immersed in a Stokes flow. The active particles transfer chemical energy into mechanical work, leading to stresses that are communicated across the system. Furthermore, particle coordination causes complex behavior inside the flow. More details are provided in (Ohana et al., 2024) for the dataset generation setup. We provide a visualization of a trajectory in fig. 11.

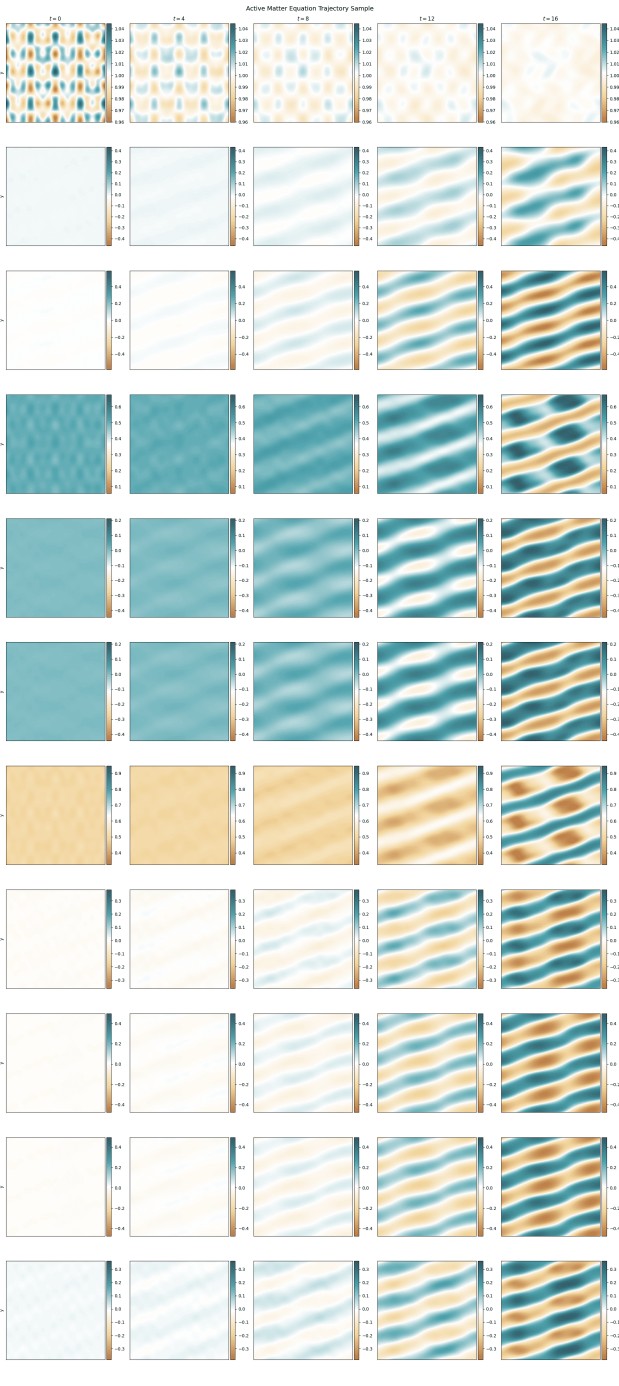

Figure 11: Trajectory sample of *Active Matter* dataset.

## B.8 ACOUSTIC SCATTERING MAZE

**Add** We use the *Acoustic Scattering Maze* dataset introduced by Ohana et al. (2024). The dynamics are governed by acoustic equations that describe the propagation of a pressure wave through materials with spatially varying density. The governing system is:

$$\frac{\partial p}{\partial t} + K(x,y)\left(\frac{\partial u}{\partial x} + \frac{\partial v}{\partial y}\right) = 0 \tag{23}$$

$$\frac{\partial u}{\partial t} + \frac{1}{\rho(x,y)}\frac{\partial p}{\partial x} = 0 \tag{24}$$

$$\frac{\partial v}{\partial t} + \frac{1}{\rho(x,y)}\frac{\partial p}{\partial y} = 0 \tag{25}$$

where $\rho$ denotes the material density, $u$ and $v$ the velocity components in the $x$ and $y$ directions, $p$ the pressure, and $K$ the bulk modulus. Together, $\rho$ and $K$ determine the speed of sound; in these simulations, $\rho$ varies spatially while $K$ is fixed to a constant value of $4$.

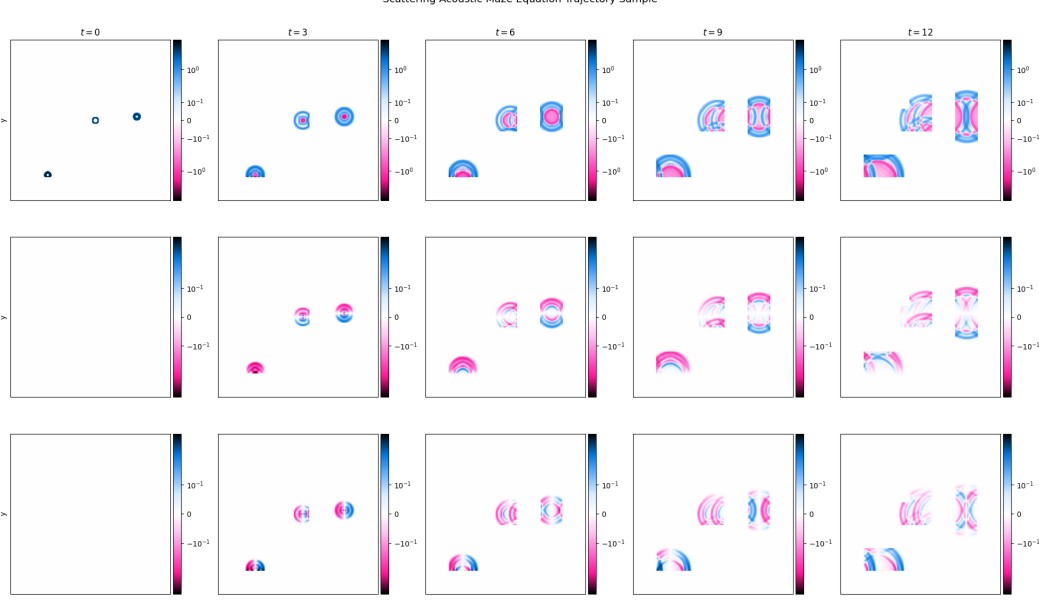

Figure 12: Sample of the *Acoustic Scattering Maze* dataset.

## B.9 MAGNETOHYDRODYNAMIC (MHD)

The MHD dataset (Ohana et al., 2024), simulates a magnetohydrodynamic (MHD) turbulence. Such dynamics are often used to describe space events such as solar winds or galaxy formation. We refer to (Ohana et al., 2024) for additional details. We provide in fig. 13 a trajectory sample.

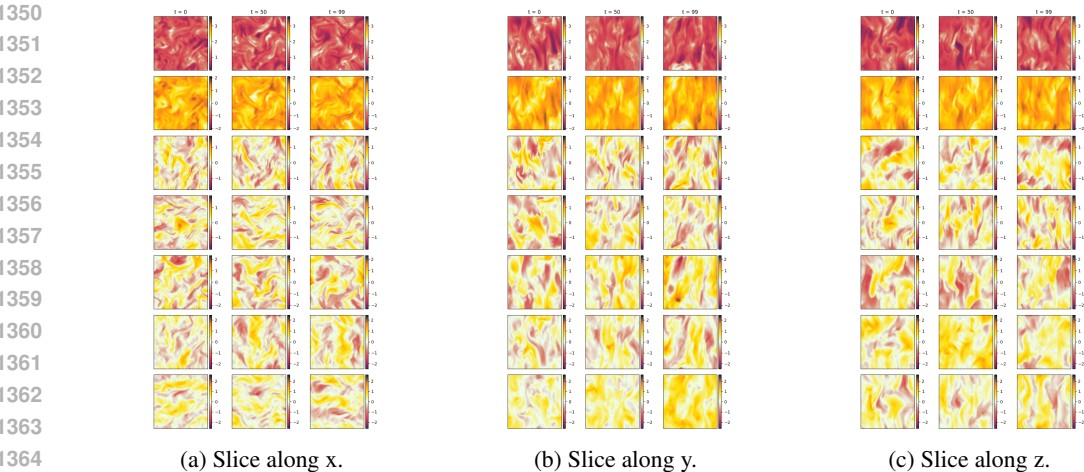

(a) Slice along x.  (b) Slice along y.  (c) Slice along z.

Figure 13: Trajectory sample of *MHD* dataset sliced along each axis.

### B.10  TURBULENCE GRAVITY COOLING (TGC)

The TGC dataset (Ohana et al., 2024), simulates a turbulent fluid with gravity. We provide in fig. 14 a trajectory sample. Additional details can be found in (Ohana et al., 2024).

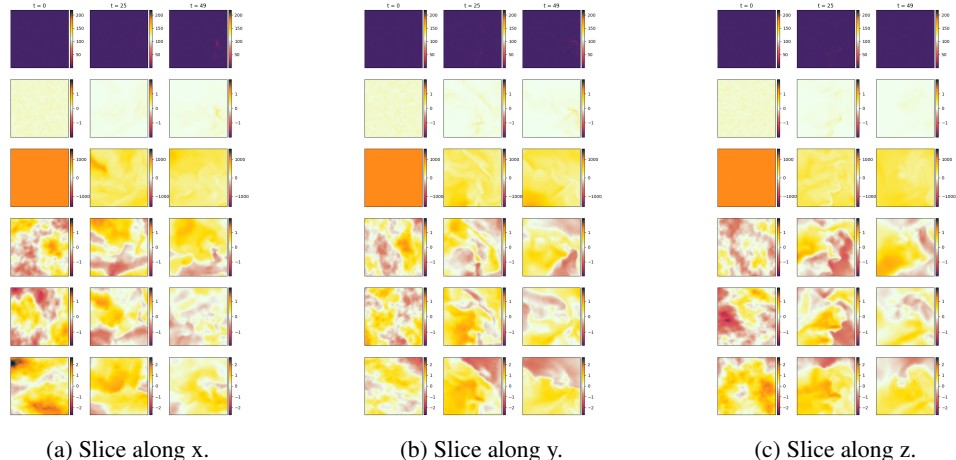

(a) Slice along x.  (b) Slice along y.  (c) Slice along z.

Figure 14: Trajectory sample of *TGC* dataset sliced along each axis.

# C ARCHITECTURE DETAILS

We provide additional architecture details about ECHO's auto encoder in appendix C.2 and generative model section 3.1.

## C.1 INFERENCE MODEL

We describe here several instance of the inference process corresponding to different tasks, that can be handled by ECHO. This is an illustration of the inference formalism introduces in section 2.1.

- **Forward prediction:** $\mathcal{O}$ contains the first $L$ frames of a trajectory, encoded as $\boldsymbol{z}^{0:L'-1} = E_\phi(\boldsymbol{u}^{0:L-1}) \in \mathbb{R}^{M \times L' \times d}$. The model predicts the missing tokens $\boldsymbol{z}^{\mathcal{M}'} = \{\boldsymbol{z}^{L'}, \ldots, \boldsymbol{z}^{T'}\}$, completing the latent trajectory.
- **Interpolation:** $\mathcal{O}$ consists of $L$ non-consecutive frames $\{\boldsymbol{u}^{t_1}, \ldots, \boldsymbol{u}^{t_L}\}$ with $\{t_0, \ldots, t_{L-1}\} \subset [0, T]$. Each observed frame is encoded spatially as $\boldsymbol{z}^{t_\ell} = E_\phi(\boldsymbol{u}^{t_\ell}) \in \mathbb{R}^{M \times d}$, yielding a set of latent observations $\boldsymbol{z}^{\mathcal{O}'}$. The generative model then reconstructs the missing temporal dynamics by producing the full latent trajectory $\boldsymbol{z}^{0:T'}$, thus interpolating the unobserved tokens $\boldsymbol{z}^{\mathcal{M}'} = \boldsymbol{z}^{0:T'} \setminus \boldsymbol{z}^{\mathcal{O}'}$.
- **Inverse prediction:** $\mathcal{O}$ contains the last frames of a trajectory $\{\boldsymbol{u}^{t_{T-L+1}}, \ldots, \boldsymbol{u}^{t_T}\}$, encoded as $\boldsymbol{z}^{\mathcal{O}'}$. The model reconstructs earlier dynamics by generating $\boldsymbol{z}^{0:T'}$, recovering $\boldsymbol{z}^{\mathcal{M}'}$.
- **Initial value problem:** $\mathcal{O}$ contains only $\boldsymbol{u}^0$ (optionally with $\boldsymbol{\gamma}$), encoded as $\boldsymbol{z}^0 \in \mathbb{R}^{M \times 1 \times d}$. The model then generates $\boldsymbol{z}^{1:T'} = \boldsymbol{z}^{\mathcal{M}'}$, yielding the full trajectory $\boldsymbol{z}^{0:T'}$.
- **Conditional / unconditional generation:** $\mathcal{O} = \varnothing$. The model initializes $\boldsymbol{z}_0^{\mathcal{M}'} \sim \mathcal{N}(0, I) \in \mathbb{R}^{M \times T' \times d}$ and generates $\boldsymbol{z}^{0:T'}$, either unconditionally or conditioned on PDE parameters $\boldsymbol{\gamma}$.

**Spatial tasks** ECHO's encoder–decoder architecture naturally supports a variety of spatial tasks. Thanks to the continuous convolution layers, the model can encode an arbitrary number of input points and decode at any desired spatial location. As a result, ECHO can perform spatial interpolation, super-resolution, and related tasks simply by querying the decoder at the appropriate target points.

- **Spatial interpolation**: $\Xi$ denotes an incomplete irregular grid. During decoding, we query the decoder at the missing locations, effectively interpolating the field on the unobserved points.
- **Super-resolution**: During decoding, we query a finer grid that includes intermediate locations between the original points, thereby interpolating the input data at a higher spatial resolution.

We provide examples of super-resolution in table 4 and spatial interpolation (reconstruction from irregular partial grids) in tables 3 and 18.

## C.2 ENCODER-DECODER

The encoder–decoder architecture consists of two stages. First, a continuous convolution layer regularizes the (potentially) irregular input grid (appendix C.2.1). Second, a spatio-temporal CNN compresses the regularized representation to the desired token dimension (appendix C.2.2), as illustrated in fig. 15.

### C.2.1 INTERPOLATION WITH CONTINUOUS CONVOLUTION

When encoding (possibly) irregular high-dimensional data, the computational burden falls on the encoder, which must process a large number of spatial points. The first layer plays a crucial role in reconstruction performance: applying a very high compression ratio leads to information loss, whereas avoiding compression results in significant computational cost (as detailed in section 4). In ECHO, we adopt an architecture (Hagnberger et al., 2025) that leverages continuous convolutions to handle irregular grids.

We now detail the continuous convolution layers used in the encoder and decoder.

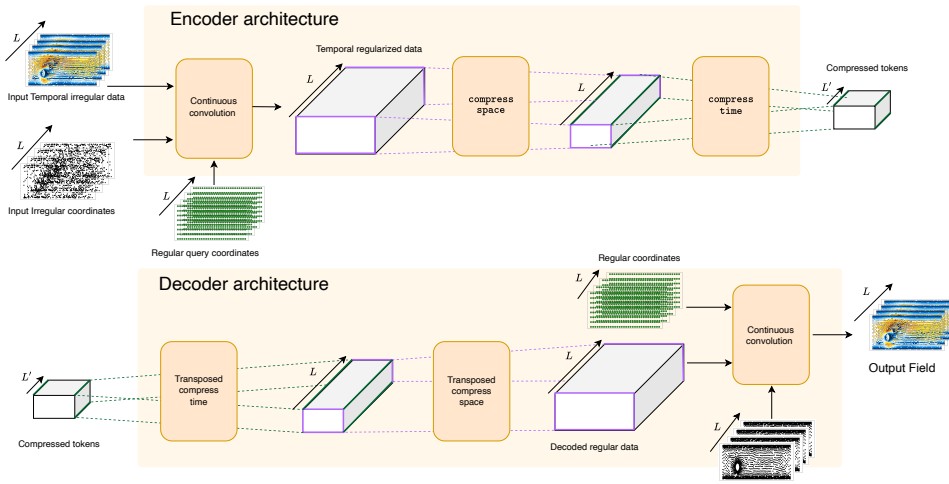

Figure 15: Encoder–decoder architecture of ECHO. The auto-encoder uses a continuous convolution layer to process irregular input grids with an arbitrary number of points, mapping the dynamics onto a regular latent grid. This latent representation is then compressed by a hierarchical CNN encoder. The decoder mirrors this process with a hierarchical CNN decoder followed by a continuous convolution layer, enabling queries at any desired output location.

**Continuous convolution for the encoder.** In the encoder, the inputs are: (i) a physical field with $C_{\text{in}}$ channels, $f^{\mathcal{O}}(\mathcal{X})$, defined on coordinates $\mathcal{X} = \{x_p\}_{p=1}^{|\mathcal{X}|}$ (with $|\mathcal{X}|$ the number of spatial points), and (ii) a regular latent grid $\Xi = \{\xi_j\}_{j=1}^{S}$, where $S$ is the number of latent grid points.

With these notations, the continuous convolution at an output location $\xi_j \in \Xi$ is given by

$$(f^{\mathcal{O}} * k)_o(\xi_j) = \sum_{i=1}^{C_{\text{in}}} \sum_{p \in \text{RF}(\xi_j)} f_i(x_p)\, k_{i,o}(\xi_j - x_p), \tag{26}$$

where $\text{RF}(\xi_j)$ denotes the receptive field around $\xi_j$, and $k_{i,o}$ is a kernel parameterized by a neural network. Each output on the regular latent grid is thus obtained by convolving the input field with this learned kernel. Importantly, there is no constraint on the number of input points $|\mathcal{X}|$, so the encoder can handle arbitrary point clouds.

**Continuous convolution for the decoder.** The final decoder layer has a similar form. Let $f^{\text{DEC}}$ be the decoded latent representation with $d$ channels, defined on the regular grid $\Xi$ (the same latent grid as in the encoder). Given a set of query coordinates $\mathcal{X} = \{x_i\}$ in physical space, the reconstructed field at a query point $x_i$ is computed as

$$(f^{\text{DEC}} * k)_o(x_i) = \sum_{c=1}^{d} \sum_{p \in \text{RF}(x_i)} f_c^{\text{DEC}}(\xi_p)\, k_{c,o}(x_i - \xi_p), \tag{27}$$

where $k_{c,o}$ is again a learned kernel and $\text{RF}(x_i)$ denotes the receptive field around the query location $x_i$.

Because the convolution kernel $k$ is learned during training and the formulation is defined for arbitrary query coordinates, the model can be evaluated at any location $x_i$. This enables a range of spatial tasks such as spatial interpolation and super-resolution, as demonstrated in section 4 and appendix E.

**ECHO's architectural modifications** In order to improve the scalability of this layer, we perform some architectural modifications.

- **Fixed latent grid**: The latent grid is fixed to a dense regular grid (up to 36k points for *Vorticity*). Contrary to (Hagnberger et al., 2025), we do not learn the grid during training,

but rather strongly enforce the latent grid to be regular. This leads to a lower memory consumption at training. This is also motivated by the use of CNN-based compression layers that needs to operate on regular grids and enables efficient and scalable encoding.

- **Chunk**: We perform chunking for very dense grids, i.e. we compute the output locally by iterative on smaller amount of query points.

### C.2.2 SPATIAL AND TEMPORAL COMPRESSION

For the architecture of the CNN, we adopt a design similar to that of Yu et al. (2023); Koupaï et al. (2025). The following description follows the one proposed in (Koupaï et al., 2025). We refer the reader to the associated article for additional details. In our case, the encoder CNN takes as input the output of the interpolation module detailed above, ie a regular representation of the input field. The regularized physical space tensor is then compressed through a stack of three building blocks: `residual`, `compress_space`, and `compress_time`. Finally, a last layer projects the representation back to the target token dimension. The decoder mirrors this compression pipeline. This type of architecture has been shown to be effective for spatio-temporal compression (Serrano et al., 2025; Yu et al., 2023; Koupaï et al., 2025). The three types of layers used in the compression module are detailed below (see (Koupaï et al., 2025)).

**`residual` blocks:** The `residual` block processes the input while preserving its original shape. It consists of a causal convolution with kernel size $k$, followed by a linear layer and a Global Context layer adapted from Cao et al. (2023). If the output dimensionality differs from the input's, an additional convolution is used to project the spatial channels to the desired size.

**`compress_space` blocks:** The `compress_space` block reduces spatial resolution by a factor of $2^d$, i.e., each spatial dimension is downsampled by a factor of 2 using a convolutional layer with stride $s = 2$. The kernel size $k$ and padding $p$ are set accordingly, with $p = k//2$. To ensure that only spatial dimensions are compressed, inputs are reshaped so that this operation does not affect the temporal axis.

**`compress_time` blocks:** The `compress_time` block performs temporal compression similarly to the spatial case, but operates along the time dimension. To preserve causality, padding is applied only to the past, with size $p = k - 1$, so that a frame at time $t$ attends only to frames at times $< t$. A convolution with stride $s = 2$ is used to reduce the temporal resolution by a factor of 2.

**ECHO's architectural choices** In our experiments, we compress physical fields by separating spatial and temporal compression. We start by stacking several `compress_space` blocks to reduce the spatial dimensions of the tokens. Then, we process it using a `residual` layer. Since these layers, make use of 3D causal convolution, they allow to process jointly the spatial and temporal dimensions. Finally, we compress the temporal dimensions with `compress_time` layers. This gives us an encoded representation of the input physics. The detailed number of layers and hyper-parameters related to the encoder/decoder modules are shown in table 11.

# D IMPLEMENTATION DETAILS

The code has been written in Pytorch (Paszke et al., 2019). All experiments were conducted on a H100. We estimate the total compute budget—including development and evaluation—to be approximately 1000 GPU-days.

## D.1 EVALUATION PROTOCOL OF THE PROCESS

### D.1.1 BASELINE DETAILS

We detail the baseline architectures used to evaluate ECHO in both forward and inverse tasks, corresponding to experiments in section 4.2. All baselines follow a comparable training protocol (see appendix D.1.2). For forward and inverse experiments, we assume 4 known timesteps and autoregressively predict the remainder, except for ECHO (and its deterministic variant), which generates the full trajectory in a single forward pass.

**FNO.** For the Fourier Neural Operator (FNO) (Li et al., 2021), we followed the authors' recommendations and concatenated temporal history directly to the input channels. We used 10 Fourier modes in both 1D and 2D, a channel width of 128, and stacked 4 spectral layers.

**BCAT.** BCAT (Liu et al., 2025) is a deterministic block-wise causal transformer for spatio-temporal dynamics, originally designed for multi-physics problems. We adapt it to parametric PDEs. BCAT performs autoregression in physical space and uses spatial patches to reduce token count, similar to Vision Transformers (Dosovitskiy et al., 2021).

**AViT.** The Axial Vision Transformer (AViT) (Müller et al., 2022) applies attention separately along spatial and temporal dimensions. We use the same configuration as BCAT (Table 7), which we found to perform best on our PDE datasets.

Table 7: BCAT hyperparameters used across all datasets.

| Hyperparameter | Value |
| --- | --- |
| Patch size | 8 |
| Transformer depth | 6 |
| Hidden size | 512 |
| MLP ratio | 2 |
| Number of heads | 8 |
| QK normalization | True |
| Normalization type | RMS |
| Activation | SwiGLU |
| Positional embedding | Sinusoidal |

**Transolver++.** Transolver++ (Luo et al., 2025) extends (Tran et al., 2023) for dense input grid, with improved parameterization and efficiency. We followed the recommended setup summarized below, provided in the reference paper.

Table 8: Transolver++ hyperparameters used across all datasets.

| Hyperparameter | Value |
| --- | --- |
| Number of layers | 8 |
| Hidden size | 128 |
| Attention heads | 8 |
| MLP ratio | 4 |
| Dropout | 0 |
| Physics slices | 32 |
| Activation | GELU |

**ENMA.** For ENMA (Koupaï et al., 2025), we implemented the generation architecture given in the original paper. We followed the authors recommendation for the hyper-parameters. Its configuration across datasets is summarized in table 9.

Table 9: ENMA generation hyperparameters.

| Hyperparameter | Active Matter | Rayleigh-Bénard | Gray-Scott | ASM |
|---|---|---|---|---|
| VAE embedding dim | 4 | 4 | 4 | 8 |
| Number of tokens | 16 | 16 | 64 | 256 |
| Patch size | 2 | 2 | 1 | 2 |
| Spatial Transformer depth | 8 | 8 | 8 | 8 |
| Causal Transformer depth | 8 | 8 | 8 | 8 |
| Hidden size | 512 | 512 | 512 | 512 |
| MLP ratio | 2 | 2 | 2 | 2 |
| Attention heads | 8 | 8 | 8 | 8 |
| Dropout | 0 | 0 | 0 | 0 |
| QK normalization | True | True | True | True |
| Normalization type | RMS | RMS | RMS | RMS |
| Activation | SwiGLU | SwiGLU | SwiGLU | SwiGLU |
| Positional embedding | Sinusoidal | Sinusoidal | Sinusoidal | Sinusoidal |
| FM steps | 10 | 10 | 10 | 10 |

### D.1.2 TRAINING DETAILS

All models were trained under the same protocol for fair comparison. Unless specified, we used the AdamW optimizer with $\beta_1 = 0.9$, $\beta_2 = 0.95$, a cosine learning rate schedule from $10^{-3}$ to $10^{-5}$, and linear warm-up over the first 2000 steps. Baselines were trained with the same schedule and selection criterion as ECHO, using the best checkpoint on train error.

Table 10: Training hyperparameters used across datasets.

| Hyperparameter | Value |
|---|---|
| Epochs | 200 |
| Batch size | 32 |
| Learning rate | $10^{-3}$ (cosine decay) |
| Weight decay | $10^{-4}$ |
| Grad clip norm | 1 |
| Betas $(\beta_1, \beta_2)$ | (0.9, 0.95) |

## D.2 ENCODER-DECODER IMPLEMENTATION PROTOCOL

### D.2.1 ECHO ENCODER-DECODER'S ARCHITECTURE HYPER-PARAMETERS

We present in table 11 the hyper-parameters used for the experiments done in section 4.3 for our encoder-decoder module.

Table 11: Hyper-parameters details of the encoder and decoder components of ECHO.

| Module | Block | Parameter | Vorticity | Shallow-Water | Eagle | Cylinder Flow |
|---|---|---|---|---|---|---|
| | | token dim | 32 | 32 | 32 | 32 |
| Interpolation module | | Regularized grid | $192 \times 192$ | $64 \times 128 \times 1$ | $64 \times 32$ | $64 \times 16$ |
| | | hidden dim | 64 | 64 | 64 | 64 |
| | | receptive field | 0.01 | 0.01 | 0.01 | 0.01 |
| | | softmax temp | 1 | 1 | 1 | 1 |
| | | chunk size | 4096 | 4096 | 4096 | 4096 |
| | | max neighbors | 512 | 512 | 512 | 16 |
| | | latent grid type | euclidian | spherical | euclidian | euclidiant |
| | | spatial dim | 2 | 2 | 3 | 2 |
| Compression module | Compression layers | Spatial compression layers | 3 | 2 | 2 | 2 |
| | | Temporal compression layers | 1 | 2 | 2 | 2 |
| | | Residual layers | One between each compression layer | | | |
| | | compression kernel size | 3 | 3 | 3 | 3 |
| | causal input layer | kernel size | 3 | 3 | 3 | 3 |
| | causal output layer | kernel size | 3 | 3 | 3 | 3 |

### D.2.2 BASELINE DETAILS

We detail here the hyper-parameters used for the baselines considered in section 4.3. If not explicitly stated, we follow the authors recommendation in the respective papers. In table 12, we report the shapes of the trajectory in the physical space and in the latent space for each baseline. GINO, CORAL, AROMA and CALM-PDE only compress information spatially. In ECHO, both temporal and spatial space can be compressed.

Table 12: Physical and latent space considered in experiments presented in table 3. We use the format: temporal size × spatial size × token dimension. *Trajectory sizes* report the shape of the trajectories considered. For datasets with a high number of input points, some baselines report a smaller latent shape (reported in red) to fit the memory of the GPU.

| Model | Vorticity | Shallow Water | Eagle | Cylinder Flow |
|---|---|---|---|---|
| *Trajectory sizes* | $10 \times (512 \times 512) \times 1$ | $10 \times (128 \times 256) \times 2$ | $200 \times 3000 \times 3$ | $60 \times 2000 \times 3$ |
| GINO | $10 \times 576 \times 32$ | $40 \times 512 \times 32$ | $100 \times 512 \times 32$ | $60 \times 64 \times 32$ |
| CORAL | $10 \times 18432$ | $40 \times 16384$ | $200 \times 4096$ | $50 \times 2048$ |
| AROMA | $10 \times 64 \times 32$ | $40 \times 128 \times 32$ | $200 \times 128 \times 32$ | $60 \times 64 \times 32$ |
| CALM-PDE | $10 \times 576 \times 32$ | $40 \times 512 \times 32$ | $200 \times 128 \times 32$ | $60 \times 64 \times 32$ |
| ECHO | $5 \times 24 \times 24 \times 32$ | $10 \times 16 \times 8 \times 32$ | $50 \times 16 \times 8 \times 32$ | $15 \times 16 \times 4 \times 32$ |

**GINO** In GINO, a graph operator is proposed relying on a GNO block to encode mesh points into a latent space. The resulting latents are then processed by 8 FNO layers to increase model expressivity. We use the default hyperparameters provided in the `neuralop` package (Kossaifi et al., 2025; Kovachki et al., 2023).

**CORAL** CORAL is a operator learning framework relying on cordinate-based networks for solving PDEs on general geometries. The INRs are learned via a second-order CAVIA-style meta-learning scheme that optimizes shared parameters so that, for each new function, only a few inner-loop gradient steps are required to obtain an accurate per-sample latent code from sparse observations. We use 3 inner-loop steps with an inner learning rate of $10^{-2}$. The INR has 6 layers of width 256, and the hypernetwork has 1 hidden layer with width 256. We refer to (Serrano et al., 2023) for additional details.

**AROMA** The AROMA baseline uses cross-attention to directly compress trajectories into a latent space. We follow the configuration proposed in (Serrano et al., 2024) and set the number of latent tokens as detailed in table 12. Due to GPU memory limitations, we use fewer tokens than the other baselines on two datasets (Vorticity and Shallow-Water).

**CALM-PDE** For CALM-PDE, we follow the configuration from (Hagnberger et al., 2025), using 2 or 3 hierarchical convolutional layers where the spatial resolution is gradually reduced while the number of channels increases. For the first layer, which processes irregular grids, we select the largest latent grid that fits in GPU memory, and then apply a sequence of $\times 2$ downsampling steps until we reach the desired token resolution. As reported in table 3, GPU memory constraints forced us to use a smaller first-layer latent grid than ECHO on three datasets: 2048 for Vorticity, 1024 for Shallow-Water, and 512 for Eagle.

### D.2.3 TRAINING DETAILS

**Reconstruction performances** To ensure fairness, we trained all encoder–decoders using a unified procedure. Specifically, we optimized a relative mean squared error loss between the auto-encoder output and the original function evaluated on the full spatial grid (the $100\%$ setting).

Models are trained with the AdamW optimizer, using an initial learning rate of $10^{-3}$ that is annealed to $10^{-7}$ with a cosine schedule. Each model is trained for 20 hours, and the batch size is tuned to best utilize the available GPU memory on an NVIDIA H100 (80,GB), with the chosen values reported in table 13.

Table 13: Training batch size used across datasets.

| Batch size | Task | Vorticity | Shallow-Water | Eagle | Cylnder Flow |
|---|---|---|---|---|---|
| GINO | *Rec* | 8 | 16 | 32 | 32 |
| | *TS* | 4 | 16 | 8 | 16 |
| CORAL | *Rec* | 1 | 2 | 16 | 32 |
| | *TS* | 1 | 4 | 16 | 16 |
| AROMA | *Rec* | 1 | 1 | 4 | 32 |
| | *TS* | 1 | 1 | 4 | 16 |
| ENMA | *Rec* | 1 | 1 | 1 | 8 |
| | *TS* | 1 | 2 | 4 | 16 |
| CALM-PDE | *Rec* | 1 | 1 | 2 | 2 |
| | *TS* | 8 | 16 | 8 | 16 |

For ECHO, as we do spatio-temporal encoding, we divide our training into two stages, as explained in section 3.2. We report the hyper-parameters considered in table 14.

Table 14: Training hyperparameters used across datasets for the low- and high-resolution stages.

| Hyperparameter | Low Resolution | High Resolution |
|---|---|---|
| Epochs | 200 | 10 |
| Batch size | 8 | 8 |
| Learning rate | $10^{-3}$ (cosine decay) | $10^{-4}$ (cosine decay) |
| Weight decay | $10^{-4}$ | $10^{-4}$ |
| Grad clip norm | 1 | 1 |
| Betas $(\beta_1, \beta_2)$ | (0.9, 0.95) | (0.9, 0.95) |

**Generative process training**   All models are evaluated on the forward task, where 4 time-steps are considered as context go generate the trajectory. Hyper-parameters are ported in table 15 across all baselines. For this training, we considered a maximum budget of 20 hours.

Table 15: Training hyperparameters used across datasets for the generative stage.

| Hyperparameter | Generative |
|---|---|
| Epochs | 200 |
| Batch size | 32 |
| Learning rate | $10^{-3}$ (cosine decay) |
| Weight decay | $10^{-4}$ |
| Grad clip norm | 1 |
| Betas $(\beta_1, \beta_2)$ | (0.9, 0.95) |

# E ADDITIONAL EXPERIMENTS

To further assess ECHO's capabilities beyond the main-text experiments, we report a series of additional studies covering long-range extrapolation, unconditional generation, robustness to sparse and irregular observations, and key architectural design choices. Specifically, we consider:

- **Long-range prediction** on Shallow-Water, where ECHO is evaluated up to $4\times$ its training horizon (appendix E.1). We also provide a spectra analysis on the Vorticity equation;
- **Generative trajectory sampling** in an unconditional setting on Shallow-Water, including Fréchet Physics Distance (FPD) and efficiency metrics (appendix E.2);
- **Initial value problem (IVP) evaluation**, where ECHO generates full trajectories from a single initial frame and PDE parameters (appendix E.3);
- **Temporal interpolation tasks**, in which ECHO reconstructs long missing segments between observed frames across four datasets (appendix E.4);
- **Encoding sparse data / spatial interpolation**, where only $50\%$ or $10\%$ of the input grid is observed and the full grid must be reconstructed, with $10\%$ constituting a strong OOD sparsity level (appendix E.5).

In addition, we provide a set of ablation studies in appendix E.6.1, analyzing:

- the impact of *hierarchical compression depth* in the encoder on reconstruction accuracy;
- the contribution of the *refinement stage* for full-resolution Vorticity reconstruction;
- the robustness of the *flow-matching generative process* to the choice of ODE solver and number of integration steps at sampling time.

## E.1 LONG-RANGE PREDICTION

### E.1.1 LONG-RANGE PREDICTION ON SHALLOW-WATER

To further assess the ability of ECHO to perform long-range prediction, we extend the experiments of section 4.4.1 to the Shallow-Water equation. On this dataset, ECHO is trained on trajectories of length $T = 40$ time steps, and evaluated in an out-of-distribution regime up to $4T = 160$.

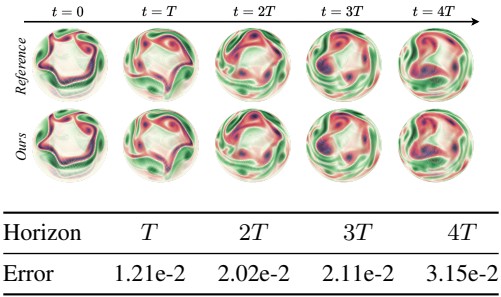

| Horizon | $T$ | $2T$ | $3T$ | $4T$ |
|---------|-----|------|------|------|
| Error | 1.21e-2 | 2.02e-2 | 2.11e-2 | 3.15e-2 |

Figure 16: Shallow-Water long-range prediction with ECHO. **Top:** rollouts over increasing horizons, starting from the same initial condition. **Bottom:** relative MSE at horizons $T$, $2T$, $3T$, and $4T$; ECHO is trained on sequences of length $T = 40$ and evaluated up to $4T = 160$ steps.

**Result** Figure 16 shows that the error increases smoothly with the rollout horizon, from $1.21 \times 10^{-2}$ at $T$ to $3.15 \times 10^{-2}$ at $4T$. This moderate and monotonic degradation indicates that ECHO produces stable long-range forecasts even when extrapolating four times beyond its training horizon.

### E.1.2 ADDITIONAL ANALYSIS ON THE VORTICITY GENERATION

While relative MSE is a standard metric in the PDE literature, it can hide over-smoothing, especially on turbulent fields where high-frequency features are important.

To better assess ECHO's ability to preserve small-scale structures, we perform a spectral analysis on the *Vorticity* dataset, both within the in-distribution training horizon and far beyond it. This type of analysis is standard in fluid dynamics and characterizes how energy is distributed across spatial scales, from large eddies at low wavenumbers $k$ to fine-scale structures at high $k$.

We proceed as follows: (i) we transform each physical field $u(x)$ into the frequency domain via FFT, obtaining Fourier coefficients $\mathcal{F}(u)(k)$ at accessible wavenumbers $k$; (ii) we compute the corresponding energy spectrum $E(k) = \frac{1}{2}|\mathcal{F}(u)(k)|^2$; (iii) we average the energy over all wavevectors with the same magnitude $|k|$ to obtain an isotropic spectrum.

The resulting spectra are shown in fig. 17 (done for the forward task): generation within the training horizon (up to $T = 20$, left) and long-range generation beyond the training horizon (up to $T = 50$, right). Each curve reports the energy spectrum averaged over all samples and time-steps in the corresponding temporal range.

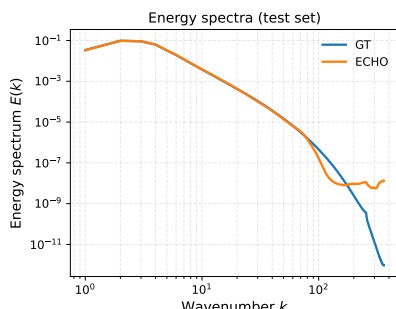 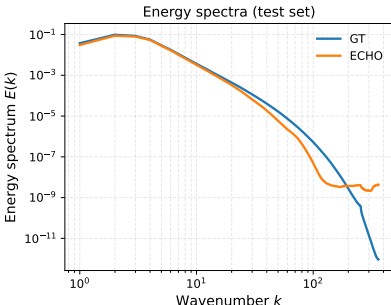

(a) Spectral analysis for generation of a trajectory inside the training horizon ($T = 20$).

(b) Spectral analysis for generation of a trajectory beyond the training horizon ($T = 50$).

Figure 17: Spectral analysis on a trajectory generation with ECHO on the *Vorticity* dataset.

**Result**   ECHO faithfully reconstructs the physics of the system across the most energy-dominant scales. The only discrepancy is a slight overestimation of energy at the very finest scales (high spatial frequencies), where most neural surrogates are known to fail. For a simulation running beyond its training horizon, ECHO shows good stability (the large scales are accurate). Beyond the training distribution, ECHO succeeds to preserve the low and medium scales, but fail to capture high and very-high spatial frequencies.

### E.2    GENERATIVE DATA SAMPLING

As a demonstration of **ECHO**'s generative capability, we assess its ability to produce diverse and physically plausible trajectories (on Shallow-Water equation) and compare its efficiency to existing neural surrogates. We evaluate its generative quality using the **Fréchet Physics Distance (FPD)** (Koupaï et al., 2025), an adaptation of the Fréchet Inception Distance widely used in image synthesis. FPD is computed in a compact 64-dimensional feature space extracted by a lightweight CNN encoder trained on physical trajectories, enabling semantically meaningful comparisons while avoiding the curse of dimensionality. Lower FPD indicates that generated trajectories are statistically closer to the training data distribution. For this evaluation, ECHO is trained in a fully *unconditional* setting, where the entire trajectory is noised and then generated.

Beyond fidelity, we report standard **efficiency metrics** relevant for large-scale PDE surrogates:

- **Throughput (Training)** — the number of samples processed per second during training (higher is better);
- **Latency** — average wall-clock time to generate a single trajectory at inference (lower is better);
- **Parameter count** — total learnable parameters, which provides a proxy for model complexity.

Table 16: **Efficiency and generative fidelity.** Throughput and latency measured on identical hardware. FPD quantifies distributional alignment with ground-truth trajectories (lower is better). Results

| Diffusion Model | Encoder/ Decoder | #Params | Throughput Training (samples/s) | Latency (s) | FPD ↓ |
|---|---|---|---|---|---|
| **Ours** | GINO | 19M | 55.7 | 1.83 | 1.68e-1 |
| | CORAL | 21M | 4.3 | 0.75 | 1.31e-1 |
| | AROMA | 0.5M | 32.8 | **0.19** | 1.55e-2 |
| | CALM-PDE | 2M | 60.5 | 0.24 | 1.02e-1 |
| | **ECHO** | **30M** | 35.0 | 0.21 | **1.12e-3** |

**Result** ECHO achieves the lowest FPD by a wide margin ($1.1 \times 10^{-3}$), indicating its generated trajectories align extremely closely with the true data distribution, significantly outperforming existing efficient surrogates such as AROMA ($1.5 \times 10^{-2}$) and CALM-PDE ($1.0 \times 10^{-1}$). While slightly heavier in parameters than minimalistic encoders, ECHO maintains competitive **throughput** (35 samples/s) and extremely low **inference latency** (0.21s/trajectory), close to the fastest baseline AROMA (0.19s) while being substantially more accurate.

### E.3 INITIAL VALUE PROBLEM

We further evaluate ECHO in an *initial value problem* (IVP) setting, where the model uses as context only the first frame of the trajectory and the governing PDE parameter (here, the viscosity) as inputs. This setting tests the model's ability to generate a full trajectory from minimal information, a common scenario in real-world PDE simulations.

We consider the **Vorticity** system on a $512 \times 512$ grid and report relative L2 reconstruction error over the full rollout. When conditioned only on the initial state and viscosity, ECHO achieves a relative L2 of $1.15 \times 10^{-1}$, demonstrating robust trajectory generation despite extremely sparse input.

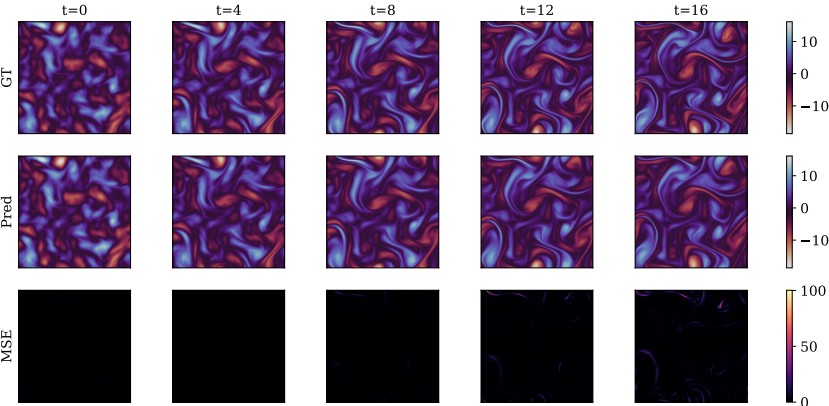

Figure 18: **Initial value problem on Vorticity** ($512 \times 512$)**.** Ground truth (GT), ECHO predictions, and per-frame error maps when generating the full trajectory from only the first state and PDE viscosity.

### E.4 INTERPOLATION TASKS

In addition to the inverse and forward settings discussed in the main text, we evaluate ECHO on a temporal interpolation task. Given the first 2 and last 2 frames of each trajectory as context, the model must reconstruct all intermediate frames (11 frames for Rayleigh–Bénard and ASM, 18 for Active Matter, and 36 for Gray–Scott). We report the Relative MSE over the interpolated segment, together with inverse and forward errors, in table 17.

Table 17: Comparison of models performance across four dynamical systems. *Determ.* = deterministic models; *Gen.* = generative models; *Inv.* = inverse (temporal conditioning) task; *For.* = forward (initial value problem) task; *Interp.* = Temporal interpolation task. All error values are Relative MSE (lower is better); "–" indicates non-convergence. Latency indicates the time for generating a whole trajectory - it is averaged here over the different datasets.

| Setting ↓ | Model ↓ | Latency (s) ↓ | Rayleigh-Benard | | | Gray-Scott | | | Active Matter | | | ASM | | |
|---|---|---|---|---|---|---|---|---|---|---|---|---|---|---|
| | | | *Inv.* | *For.* | *Interp.* | *Inv.* | *For.* | *Interp.* | *Inv.* | *For.* | *Interp.* | *Inv.* | *For.* | *Interp.* |
| Determ. | FNO | 4.42e-2 | 2.47e+3 | 4.23e-1 | | – | – | | – | – | | 1.87e0 | 1.52e0 | |
| | Trans.++ | 3.52e-1 | 6.34e-1 | 3.31e-1 | | 4.43e-1 | 2.34e-1 | | 7.33e-1 | 6.91e-1 | | 1.03e0 | 9.64e-1 | |
| | BCAT | 1.18e-1 | 1.91e-1 | 1.06e-1 | | 2.19e-1 | 8.82e-2 | | 4.98e-1 | 4.56e-1 | | 1.95e-1 | 2.18e-1 | |
| | AVIT | 1.04e-1 | 4.50e-1 | 1.01e-1 | | 1.66e-1 | 7.42e-2 | | 4.50e-1 | 4.62e-1 | | **1.04e-1** | 1.52e-1 | |
| | ECHO | 4.89e-2 | 2.53e-1 | 1.32e-1 | 1.08e-1 | 8.36e-2 | 7.66e-2 | 3.12e-2 | 5.74e-1 | 3.44e-1 | 1.47e-1 | 1.47e0 | 2.53e-1 | 1.88e-1 |
| Gen. | ENMA | 2.20e0 | 1.71e-1 | 9.87e-2 | | 1.08e-1 | 5.44e-2 | | 4.27e-1 | 3.33e-1 | | 1.01e0 | 4.12e-1 | |
| | ECHO | 1.10e-1 | **1.16e-1** | **9.28e-2** | **9.02e-2** | **2.53e-2** | **5.12e-2** | **1.78e-2** | **3.55e-1** | **2.87e-1** | **1.11e-1** | 1.12e-1 | **1.32e-1** | **1.00e-1** |

**Result** ECHO shows strong interpolation capabilities across all four systems, both for the deterministic and generative version. These results indicate that ECHO reliably fill in long missing segments between observed states.

### E.5 Encoding sparse data - Spatial interpolation

In addition to table 3, we evaluate the encoder–decoder of ECHO and all baselines under more challenging sparse-input regimes. In this experiment, the models receive only a subset of the input grid and must reconstruct the full-resolution output grid. We consider two sampling levels at test time: an intermediate sparsity of $50\%$ of the input points, and an extreme sparsity of $10\%$.

The $10\%$ setting is out-of-distribution, since all models were trained with input samplings ranging only from $25\%$ to $75\%$. This evaluation therefore probes the robustness of each method to significantly more aggressive subsampling at inference. The corresponding results are reported in table 18.

Table 18: **Reconstruction error** - Test results. Metrics in MSE. *Rec.* stands for reconstruction task, *TS* stands for Time-Stepping and *Comp.* means compression rate. Best results are **bolded** and second best are underlined. Finally, we distinguish in column *Hier.* (for Hierarchic) if the model allows for hierarchical compression (✓) or direct compression in the latent space (✗), and the strategy used for managing irregular points in column *Irr.*.

| $\mathcal{X}_{te}$ ↓ | dataset | | → | Vorticity | | Shallow-Water | | Eagle | | Cylinder Flow | |
|---|---|---|---|---|---|---|---|---|---|---|---|
| | Irr. | Model | Hier. | *Rec.* | *TS* | *Rec.* | *TS* | *Rec.* | *TS* | *Rec.* | *TS* |
| 50% | Graph | GINO | ✗ | 9.99e-1 | 1.00 | 8.69e-1 | 1.11 | 5.80e-1 | 1.88 | 7.94e-1 | 8.65e-1 |
| | INR | CORAL | ✗ | 5.14e-1 | 1.34 | 2.29e-1 | 6.88e-1 | 6.11e-1 | 1.54 | 3.04e-1 | 5.07e-1 |
| | Attention | AROMA | ✗ | 5.13e-1 | 8.43e-1 | 2.69e-2 | 4.28e-2 | 3.09e-1 | 3.34e-1 | **3.17e-2** | 2.31e-1 |
| | | ENMA | ✓ | 4.39e-1 | 4.39e-1 | 7.42e-2 | 7.53e-2 | 3.02e-1 | 3.42e-1 | 7.77e-2 | 1.09e-1 |
| | Convolution | CALM-PDE | ✓ | 2.76e-1 | 4.26e-1 | 2.61e-1 | 2.84e-1 | 9.34e-1 | 9.55e-1 | 2.50e-1 | 3.59e-1 |
| | | ECHO | ✓ | **7.32e-2** | **2.23e-1** | **1.94e-2** | **2.00e-2** | **1.88e-1** | **2.90e-1** | 4.72e-2 | **9.63e-2** |
| 10% | Graph | GINO | ✗ | 9.99e-1 | 1.00 | 8.83e-1 | 1.11 | 7.82e-1 | 1.15 | 7.98e-1 | 8.64e-1 |
| | INR | CORAL | ✗ | 5.27e-1 | 1.33 | 2.31e-1 | 6.86e-1 | 6.66e-1 | 1.53 | 3.77e-1 | 5.19e-1 |
| | Attention | AROMA | ✗ | 5.16e-1 | 8.44e-1 | 4.89e-2 | 5.43e-1 | 5.26e-1 | 1.01 | **7.12e-2** | 2.36e-1 |
| | | ENMA | ✓ | 4.45e-1 | 7.78e-1 | 9.35e-2 | 8.93e-2 | 4.72e-1 | 6.86e-1 | 1.10e-1 | 1.44e-1 |
| | Convolution | CALM-PDE | ✓ | **3.24e-1** | **4.59e-1** | 3.03e-1 | 3.17e-1 | 9.79e-1 | 1.02 | 3.08e-1 | 4.14e-1 |
| | | ECHO | ✓ | 5.60e-1 | 9.86e-1 | 1.06e-1 | **8.69e-2** | **3.82e-1** | **7.31e-1** | 1.13e-1 | **1.41e-1** |

**Result** Overall, table 18 shows that convolution-based hierarchical models (CALM-PDE and ECHO) outperform graph- and INR-based baselines both for reconstruction and time-stepping. At $50\%$ sampling (in-distribution), ECHO achieves the best or second-best performance across most datasets. In the more challenging $10\%$ sampling regime, which constitutes a strong OOD setting relative to the training samplings, all models degrade, but ECHO remains competitive and often matches or surpasses alternative baselines, especially on the more complex Eagle.

## E.6 ABLATION STUDIES

### E.6.1 IMPACT OF HIERARCHICAL COMPRESION

To assess the benefit of hierarchical compression in the encoder, we conduct an ablation study where the final latent space size (the bottleneck resolution) is kept fixed, but the way we reach this compressed space is varied.

A hierarchical level refers to how many successive downsampling stages are applied in space; each level reduces the spatial resolution by a factor of two along each dimension. For instance, a 3-level hierarchy means three consecutive $2\times$ spatial reductions before reaching the final latent grid, whereas level 0 means a single direct projection from the input resolution to the final latent size (no intermediate stages). Importantly, in this experiment we only compress space (not time), so the temporal dimension remains unchanged (table 19).

Table 19: Impact of hierarchical depth in the encoder on reconstruction error (RMSE). A higher level means more progressive spatial downsampling while keeping the same final latent size.

| Level | 0 | 1 | 2 | 3 |
|---|---|---|---|---|
| **RMSE** | 3.94e-1 | 3.64e-1 | 1.09e-1 | **2.87e-2** |

**Result** Table 19 shows that increasing hierarchical depth dramatically improves reconstruction accuracy while keeping the same final latent size. Without hierarchy (level 0), RMSE is high (3.94e-1), while introducing even one level of progressive compression significantly reduces error. The best performance is achieved with three levels (RMSE 2.87e-2), highlighting that iterative, deep compression yields more expressive and informative latent representations than a single aggressive projection.

This validates our design choice of a deep hierarchical encoder: even at fixed compression ratio, progressive downsampling better preserves spatial structure and leads to markedly improved fidelity.

### E.6.2 IMPACT OF THE REFINEMENT STAGE

To evaluate the contribution of the *refinement stage*, we conduct an ablation on the **Vorticity** dataset at full spatial resolution. Our training strategy first learns from spatio-temporal data that is *spatially subsampled* to reduce memory cost. After this pretraining, we apply a refinement stage that re-encodes each frame individually at *full spatial density*, improving the latent representation for high-resolution inputs.

Table 20: Effect of the refinement stage on full-grid **Vorticity** reconstruction (Relative L2, lower is better).

| | w/o refinement | w/ refinement |
|---|---|---|
| Relative L2 | 2.24e-1 | **6.88e-2** |

**Result** This additional step proves particularly beneficial for highly detailed fields such as Vorticity, where local fine-scale structures are hard to preserve under strong compression. As shown in table 20, adding the refinement stage significantly improves reconstruction accuracy.

### E.6.3 ABLATIONS ON THE GENERATIVE PROCESS

We conduct ablations on the vorticity dataset to validate the robustness of our flow-matching generative design. In particular, we vary both the ODE solver used at sampling time and the number of denoising steps, while keeping the learned model fixed.

Table 21: Ablation of the flow-matching generative process on the Vorticity dataset. We vary the ODE solver and the number of integration steps used at sampling time. Metric is Relative MSE (lower is better). Default is midpoint solver and $5$ denoising steps.

| Ablation | Configuration | Rel. MSE $\downarrow$ |
|---|---|---|
| Solver | Euler | 1.62e-1 |
| | Midpoint | 1.67e-1 |
| | RK4 | 1.68e-1 |
| | DOPRI5 | 1.71e-1 |
| # Steps | 2 steps | 1.72e-1 |
| | 5 steps | 1.69e-1 |
| | 10 steps | 1.68e-1 |
| | 25 steps | 1.67e-1 |

**Result**  Overall, the reconstruction errors remain in a very narrow range across all configurations, with differences on the order of $10^{-3}$. Simple solvers such as Euler already perform on par with higher-order methods (RK4, DOPRI5), and increasing the number of integration steps from 2 to 25 brings only marginal gains. These results indicate that our flow-matching generative process is stable with respect to the choice of ODE solver and discretization.

# F VISUALIZATIONS

We provide qualitative visualizations of ECHO's generated trajectories and a comparison with ground truth trajectories.

## F.1 VORTICITY

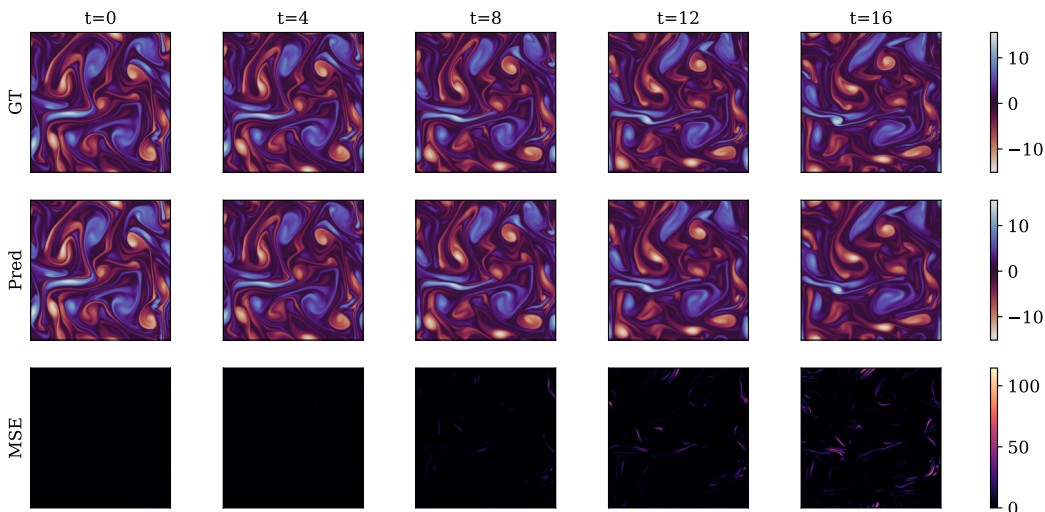

Figure 19: Generation using ECHO's framework on the Vorticity dataset. We generate $1024 \times 1024$ spatial points per frame. (Top) ground truth trajectory, (Middle) ECHO's prediction, (Bottom) Error.

## F.2 GRAY-SCOTT

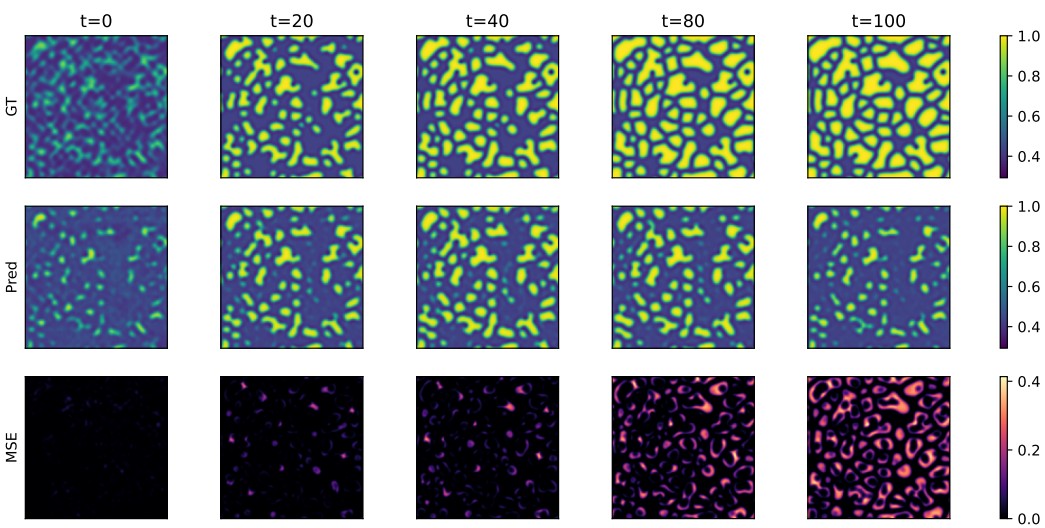

Figure 20: Generation using ECHO's framework on the Gray-Scott dataset. (Top) ground truth trajectory, (Middle) ECHO's prediction, (Bottom) Error.

### F.3 RAYLEIGH-BENARD

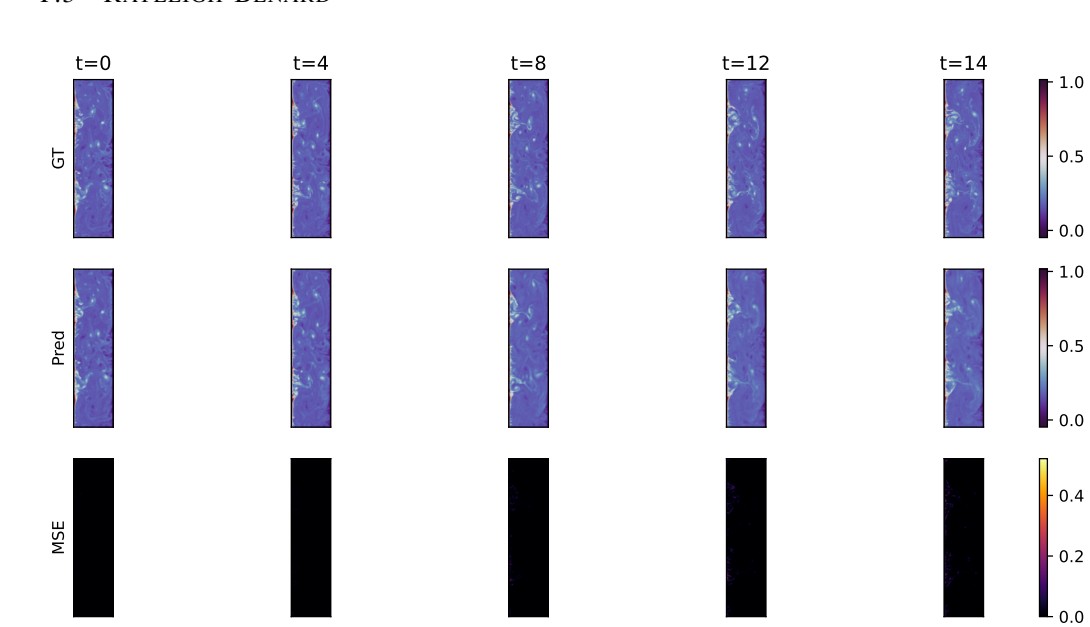

Figure 21: Generation using ECHO's framework on the Rayleigh-Benard dataset. (Top) ground truth trajectory, (Middle) ECHO's prediction, (Bottom) Error.

### F.4 ACTIVE-MATTER

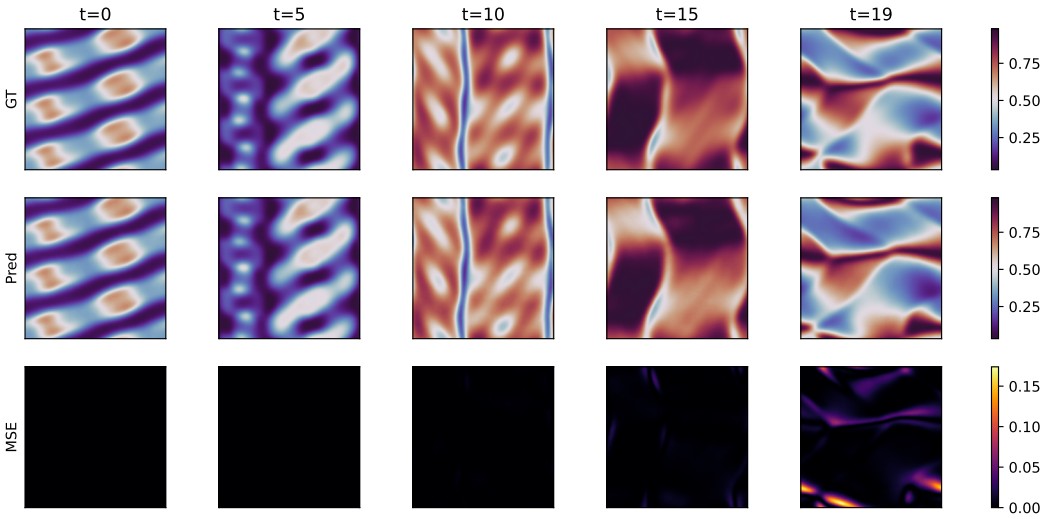

Figure 22: Generation using ECHO's framework on the Active Matter dataset. (Top) ground truth trajectory, (Middle) ECHO's prediction, (Bottom) Error.

## F.5 ACOUSTIC SCATTERING MAZE

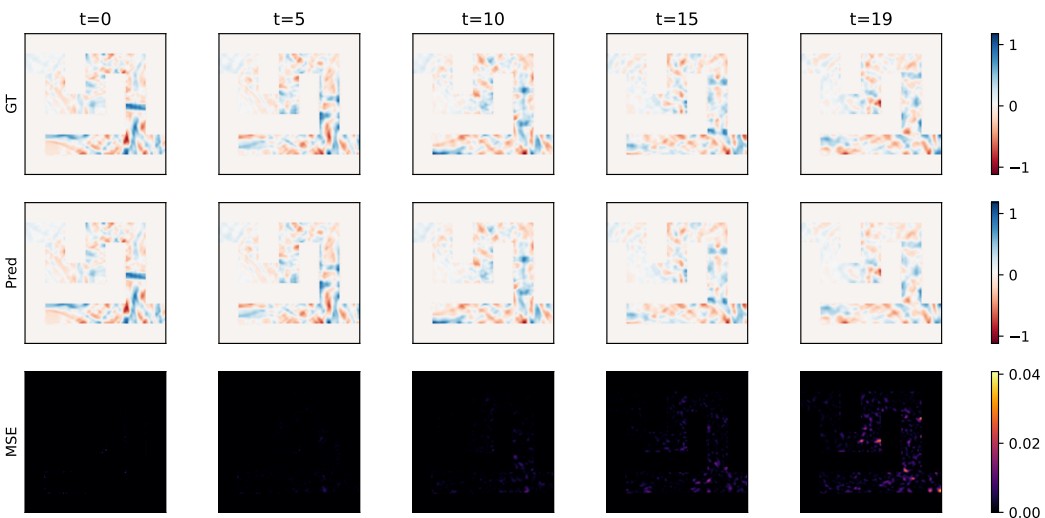

Figure 23: Generation using ECHO's framework on the Acoustic Scattering Maze dataset. (Top) ground truth trajectory, (Middle) ECHO's prediction, (Bottom) Error.

## F.6 EAGLE

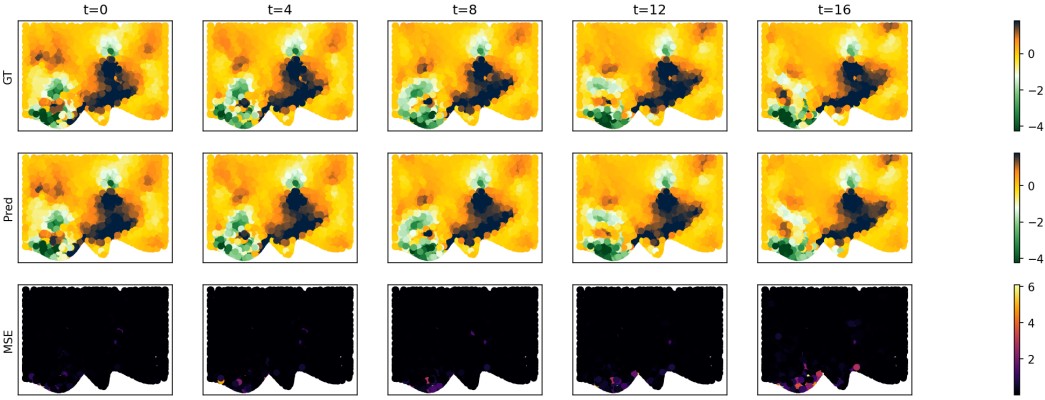

Figure 24: Generation using ECHO's framework on the Eagle dataset. (Top) ground truth trajectory, (Middle) ECHO's prediction, (Bottom) Error.

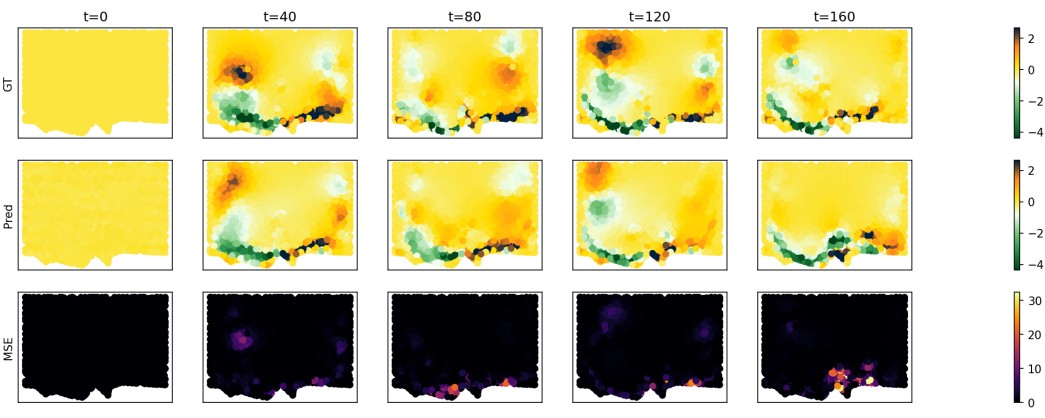

Figure 25: Generation using ECHO's framework on the Eagle dataset - long trajectory generation. (Top) ground truth trajectory, (Middle) ECHO's prediction, (Bottom) Error.

