# OpenReview forum: "Efficient Generative Transformer Operators for Million-Point PDEs"
_ICLR.cc/2026/Conference — Submitted to ICLR 2026_

### Official Review · Reviewer_uXAC · 2025-10-20

**Soundness:** 2
**Presentation:** 2
**Contribution:** 2
**Rating:** 2
**Confidence:** 4

**Summary:**

This paper presents ECHO, a transformer–operator framework for generating million-point PDE trajectories. ECHO employs a hierarchical convolutional encode–decode architecture to compress regular or irregular spatio-temporal trajectories and adopts a flow-matching model to learn the trajectory distribution. The two parts are trained separately and the generative part enables forward and inverse problems, and interpolation. The proposed method demonstrates SOTA performance on million-point simulations on complex geometries.

**Strengths:**

Overall I think the proposed framework is elegant.

- The proposed method incorporates several reasonable technical designs: a hierarchical spatio-temporal encoder-decoder which can compress the high-dimensional input efficiently; a generative flow-matching framework which is capable of modeling complex distributions and generating high-fidelity spatio-temporal dynamics. The generative modeling framework naturally enables more types of tasks as well.
- The experiments in comparison to state-of-the-art baselines seem solid. The proposed method is evaluated on two types of tasks: reconstruction tasks and time-stepping tasks. The paper also presents evidence that baseline models are not able of handling million-point inputs.
- The paper is overall easy to follow.

**Weaknesses:**

My major concerns are about the experiments and writing.

- Although the generative framework enables interpolation tasks, the corresponding experiments are not included, and therefore this capability of the model is not well evaluated.
- After carefully reading through the paper, I feel that the paper is finished in a hurry. There are several typos: Line 88 "we leverate"; Line 317 "$\gamma$if available"; Line 333 the reference is missing; Line 450 the caption only highlights the reconstruction error, but the table contains both reconstruction error and time-stepping error.
- In table 3 you compare ECHO with several baselines and show that all baselines ran OOM. However, as far as I know, at least one of your baselines in Table 2-Transolver++ supports million-scale point inputs. Why not compare with it in this task?
- The experimental design is not clearly elaborated. For the time-stepping task, what is the model input and output? Do your model generate the whole trajectory all at once or step by step? What about your baselines? The lack of explanation makes it hard to evaluate the performance.
- Why do you include the large-scale Vorticity dataset in the irregular mesh section? Isn't it regular grid?
- There should be experiments comparing the efficiency of each model, for example, the inference time of each model to generate a full trajectory.
- What is the purpose of the reconstruction task? It is not a must for models to first encode the inputs into a latent space and then decode back. For example, Transolver++. However, the comparison with it is missing in this section.

**Questions:**

Please refer to weaknesses.

**Details Of Ethics Concerns:**

No Ethics Concerns.

---

> ### Author Response · Authors · 2025-11-22
> **Answer to Reviewer uXAC**
>
> We appreciate the reviewer’s positive feedback on the clarity of our manuscript as well as the experiments. We address the raised concerns below and we provide additional experiments as requested by the reviewer, concerning the interpolation evaluation. We also performed experiments with Transolver++ and explain why it failed in our setting.
>
> # Weaknesses
>
> ## W1: Interpolation tasks
>
> Thanks for the meaningful suggestion. We provide below an evaluation of ECHO for the following interpolation task:  we used as context the first 2 frames and last 2 frames, and predict all the remaining frames (11 frames for all datasets except AM (16) and GS (36)). Since the  baselines are autoregressive, they cannot handle directly interpolation. In order to provide a comparison, we also performed an experiment with a deterministic version of ECHO. This provides a reasonable baseline since except ENMA, all the other baselines used in the paper are deterministic. The performance illustrate the ability of ECHO to perform efficient interpolation. These results are now added in appendix E4 of the updated manuscript. Additionally, please note that we provided in Fig 1 right an evaluation of ECHO on the interpolation task on the Rayleigh-Bénard dataset.
>
> |Baseline | AM | GS | ASM | RB |
> |-----|-----|-----|-----|-----|
> |ECHO deterministic| 1.47e-1 | 3.12e-2 | 1.88e-1 | 1.08e-1|
> |ECHO generative|  1.11e-1 | 1.781e-2 | 1.00e-1 | 9.02e-2|
>
> ## W2: Typo errors
>
> We would like to thanks the reviewer for identifying typos. We have carefully checked and corrected all the remaining typos in the manuscript.
>
> ## W3: Transolver++ comparison for million-point trajectories
>
> This is an excellent remark.
> - **Why we did not include Transolver++ in table 3.**
> In table 2, we compared with several baselines for the forward and inverse tasks. The baselines include both deterministic and generative methods. They either operate directly in the physical space or in a latent space.  Note that results in table 2 show that Transolver++ fails to correctly perform inverse and forward tasks, compared to other baselines.
> For the irregular mesh experiments in Section 4.3, we specifically selected baselines that operate within a compact latent space and support resolution-independent querying. As Transolver++ processes data directly in the physical space and lacks the architectural flexibility to be queried at arbitrary output resolutions, it was excluded from this specific comparison.
>
> - **Additional experiments with Transolver++ on the million point experiment.**
> Per your request, we attempted to evaluate Transolver++ in the specific setting of Table 4: predicting a full trajectory based on 4 context frames. Consistent with our other experiments, this was conducted on a single H100 GPU (80GB memory). However, this setting requires processing 4 million points for the context alone (1 million per frame) and generating over 20 million output points. This volume exceeds the memory capacity of a single H100, resulting in Out-Of-Memory (OOM) errors. While Transolver++ is effective in distributed settings, it does not scale to this single-GPU million-point regime.
>
> ## W4: Descriptions of the experiments
>
> Thanks for pointing out this. We address all your remarks in the updated version of the paper (lines 350-360, 375-382 and 416-426). We provide below for clarity a synthetic description of the experiments, the details being included now in the paper.
>
> The experiments in Table 3 (formerly table 4) are designed to assess the advantage of our processor design, which generates entire trajectory segments in one shot, over state-of-the-art baselines. We benchmarked performance on both forward tasks (conditioning on the initial 4 frames) and backward tasks (conditioning on the final 4 frames). The results, reported as Relative L2 and latency in table 2, demonstrate the robustness of our Transformer design in handling multi-tasks efficiently.
>
> In Table 3 (formerly Table 4), our objective is to benchmark state-of-the-art architectures designed to map irregular meshes into compact latent spaces. We evaluate the capacity of each encoder-decoder pair using two distinct metrics:
>
> - **Reconstruction Quality ('Rec')**: We first trained each encoder-decoder on a pure reconstruction objective and measured fidelity using Relative MSE. To ensure a fair comparison, we fixed the latent space dimensionality across all methods, noting exceptions where Out-Of-Memory (OOM) constraints necessitated size reductions (indicated with colors).
>
> - **Latent Space Expressivity ('For')**: We assessed the suitability of the learned latent spaces for dynamics modeling via a forward forecasting task. To isolate the quality of the encoding, we utilized the ECHO processor across all experiments. The processor was trained individually on the latent representations produced by each baseline, using 4 context frames to predict the remainder of the trajectory.

---

> > ### Author Response · Authors · 2025-11-22
> > **Answer to Reviewer uXAC**
> >
> > ## W5: Inclusion of Vorticity dataset in the irregular section.
> >
> > Yes, the vorticity equation is defined on a regular mesh. In this section we built an artificial irregular dataset from the original vorticity dataset by sampling points at random positions.  This is a common practice for evaluating neural operators on irregular datasets, see e.g. recent works such as [1, 2].
> >
> > [1] GridMix: Exploring Spatial Modulation for Neural Fields in PDE Modeling Download PDF Honghui Wang, Shiji Song, Gao Huang.
> >
> > [2] Armand Kassaï Koupaï, Lise Le Boudec, Louis Serrano, and Patrick Gallinari. ENMA: Tokenwise autoregression for continuous neural PDE operators, 2025.
> >
> > This also corresponds to frequently encountered real-world scenarios requiring to acquire data from noisy or faulty sensors. In such scenarios, points may become unavailable, and the grid naturally becomes irregular. In our experiments, we assume that at inference we can only observe e.g., 20% of the input points, and need to generate the remaining points. In section 4.3, we evaluate ECHO on a variety of datasets exhibiting diverse complex settings (dense number of points, irregular grid, spherical coordinates, long-range generation, …). We believe this shows the potential of ECHO to serve as a solution for a variety of spatio-temporal problems.
> >
> > ## W6: Inference time
> > Maybe this was not clear enough, but efficiency aspects are already addressed in the paper in tables 2 (latency for full trajectory generation) and Appendix E.2 (training throughput and inference latency for irregular grids). In table 2, column "Latency" corresponds to the elapsed time required to generate a whole trajectory. All the baselines are compared on this task: ECHO outperforms generative SOTA ENMA framework in terms of latency and is competitive with the fastest baseline FNO for its deterministic versions. Concerning experiments in Table 3, we provided training throughput (number of samples processed per second at training) and inference latency metrics (time required for generating a whole trajectory) in appendix E2. ECHO is competitive with all baselines, while allowing million points support and superior performance.
> >
> > ## W7: Purpose of the reconstruction task
> > The paper addresses the problem of generating whole spatio-temporal trajectories. This task is computationally demanding and it is largely agreed that an essential requirement for scaling is that time stepping shall be performed in a compact latent space. This is why many recent frameworks addressing spatio-temporal forecasting adopt this encode-process-decode framework. Indeed table 2 demonstrates that operating in a latent spaces leads to lower latencies. This is then the setup chosen here. Within this context, we compare ECHO to competitive  efficient baselines operating in latent space. By efficient here, we refer to models that:
> > 1. has low latency (time to generate one complete trajectory).
> > 2. can handle large number of input points and be queried at any input resolution
> > Transolver++ does not meet these requirements.
> >
> > We reported also the reconstruction objective because when using an encode-process-decode framework, the model’s performance will remain limited by the quality of the decoding stage. The better the reconstruction quality, the better the possibility to do accurate predictions in the physical space. The goal of these experiments is therefore to highlight the capacity of the encoders-decoders reconstruction, and the expressivity of the latent space for the Forward task.
> >
> > We have clarified this aspect in the revised version of the manuscript.
> >
> >
> > We would like to highlight that in addition to the added experiment cited below, we also added additional experiments on 3d datasets (section 4.5), we have evaluated long-range prediction on turbulent dynamics (section 4.4) and a spectral analysis on the generation (in appendix E1), and we have added ablations studies on the generative component in appendix E6 .
> > We hope this address all your concerns. We have modified the paper accordingly with all your feedbacks. We believe this improve the quality of the paper and better reflect the novelty and soundness of our work.
> > We thank the reviewer again for their valuable feedback and are happy to engage further if needed.

---

### Official Review · Reviewer_HKTM · 2025-10-27

**Soundness:** 2
**Presentation:** 3
**Contribution:** 2
**Rating:** 4
**Confidence:** 4

**Summary:**

This paper introduces ECHO, a transformer–operator framework for generating million-point PDE trajectories. It employs a hierarchical convolutional encode–decode architecture that leverages the strengths of generative models while supporting multiple tasks (commonly seen in video domains). The experiments are solid and comprehensive.

**Strengths:**

1. The chosen problem scope is broad, showing potential for ECHO to serve as a foundation model for PDE solving.
2. Extensive experiments are conducted, which appear quite thorough. The experiments at 1024-resolution are particularly impressive.
3. The two-stage encoding design is well-motivated, as standard VAEs fail to compress long trajectories at high resolution effectively.

**Weaknesses:**

1. Regarding the latent grid mapping: how does the use of continuous convolution differ from constructing a nearest-neighbor graph and applying a GNN? A clarification or comparison would strengthen the methodological justification.
2. Is the assumption of a uniform latent grid always reasonable? For example, if input points are distributed on a 3D spherical surface, a uniform grid would significantly increase the number of points to process. How does the method avoid unnecessary computations for points inside the sphere?
3. Lines 245–246: The concatenation of noise and conditions which have same shape as noise is a natural and commonly used conditioning approach, while AdaLN is relatively less common for such condition types. Presenting this as a contribution may be overstated.
4. Lines 252–254: The embedding of PDE parameters has already been discussed in UniSolver [1]. The authors should appropriately acknowledge this prior work.
5. Could the authors provide more details on the three-stage training? For example, what batch size was used, and what was the GPU memory used for every model?
6. Sampling Process: The details and rationale behind the sampling strategy are unclear. How does ECHO perform sampling (e.g., the number of sampling steps, sampler type)? What ablation studies were conducted to determine and validate this sampling strategy? The lack of this information makes it difficult to assess the efficiency and optimality of the generative process.
7. DrivAerNet++ is a relevant and challenging dataset with more realistic flow simulations around vehicles. Experiments on this dataset could further validate the method's effectiveness and generalization.

[1] Unisolver: PDE-Conditional Transformers Towards Universal Neural PDE Solvers

**Questions:**

See Weaknesses

---

> ### Author Response · Authors · 2025-11-22
> **Answer to Reviewer HKTM (1)**
>
> We appreciate the reviewer’s positive feedback on the potential of ECHO as well as the experiments. We address the raised concerns below and we provide additional experiments:
>
> # Weaknesses
>
> ## W1: continuous convolutions vs GNNs.
>
> We thank the reviewer for this extremely relevant question. Both approaches are related but they manipulate different mathematical concepts. We firmly believe that our continuous approach offers several benefits over GNNs.
>
> While GNNs are a standard tool for modeling dynamics ([1], [2]) or design optimization ([3]), they rely on nearest-neighbor message passing within a fixed radius. This approach has three significant drawbacks compared to our method:
>
> - **Mesh Bias**: GNNs tend to overfit to the specific mesh topology seen during training, which hinders generalization to irregular grids with different node densities or locations (see Table 3 in [4]).
>
> - **Memory Scalability**: Storing adjacency lists for every node becomes computationally prohibitive for the large, dense graphs required in million-point simulations.
>
> - **Spatial Querying**: Unlike operators, GNNs are node-bound and cannot be easily queried at arbitrary off-grid locations.
>
> In contrast, ECHO utilizes continuous convolutions to approximate the integral operator between two functions: the input field and a learned kernel parameterized as a neural network (see Eqs. 2–4 in [5]). This formulation allows us to learn mappings between continuous function spaces effectively, enabling the model to be queried at any spatial position within the domain regardless of the input discretization.
>
> [1] Tobias Pfaff, Meire Fortunato, Alvaro Sanchez-Gonzalez, and Peter W. Battaglia. Learning mesh-based simulation with graph networks, 2021.
>
> [2] Message passing neural pde solvers. International Conference on Learning Representations, 2022.
>
> [3] Physical Design using Differentiable Learned Simulators, Kelsey R. Allen and Tatiana Lopez-Guevara and Kimberly Stachenfeld and Alvaro Sanchez-Gonzalez and Peter Battaglia and Jessica Hamrick and Tobias Pfaff, 2022.
>
> [4] Continuous PDE Dynamics Forecasting with Implicit Neural Representations, Yuan Yin, Matthieu Kirchmeyer, Jean-Yves Franceschi, Alain Rakotomamonjy, Patrick Gallinari, 2022
>
> [5] Jan Hagnberger, Daniel Musekamp, and Mathias Niepert. CALM-PDE: Continuous and adaptive convolutions for latent space modeling of time-dependent PDEs, 2025.
>
>
> ## W2: Uniform latent grid
>
> We thank the reviewer for this opportunity to clarify our 3D spherical encoding strategy. We agree that a naive dense Euclidean grid would incur prohibitive computational costs, as a significant portion of the grid points would fall outside the physical domain (empty space). To mitigate this, ECHO employs a coordinate transformation. By aligning the latent grid with the manifold of the data — for instance, utilizing a 2D regular grid in spherical coordinates for 3D spheres — we effectively eliminate unnecessary points. This strategy significantly reduces the token count and computational burden while maintaining high resolution on the physical mesh.
>
> More generally, ECHO is designed to map mesh points from arbitrary, potentially irregular input domains onto a structured latent grid defined in an appropriate coordinate system (e.g., Euclidean, spherical, or other manifolds). This geometric flexibility allows us to handle diverse topologies effectively. We have empirically validated this capability on datasets with complex boundary conditions (Eagle) and spherical geometries (Shallow-Water). As detailed in Table 17, ECHO successfully learns these dynamics while maintaining high computational efficiency and low latency.
>
> ## W3: Noise and conditions concatenation
>
> Thanks for the remark. We have reformulated the text to moderate the claim.
>
> Although concatenation strategies exist in vision, applying them to physical dynamics on irregular meshes is non-trivial and novel. Our work represents the first application of this strategy to spatio-temporal PDE forecasting on arbitrary geometries within a latent space. Prior diffusion-based PDE solvers [1, 2, 3] have predominantly relied on adaptive layer normalization (AdaLN) for conditioning. In contrast, our framework directly concatenates noise and physical inputs in a compressed spatio-temporal latent space. This design choice is critical, as it enables the model to condition on arbitrary states, thereby facilitating zero-shot multi-task solving at inference.
>
> [1] Zijie Li, Anthony Zhou, and Amir Barati Farimani. Generative latent neural pde solver using flow
> matching, 2025.
>
> [2] Anthony Zhou, Zijie Li, Michael Schneier, John R Buchanan Jr, and Amir Barati Farimani. Text2PDE: Latent diffusion models for accessible physics simulation, 2025
>
> [3] Unisolver: PDE-Conditional Transformers Towards Universal Neural PDE Solvers. Hang Zhou, Yuezhou Ma, Haixu Wu, Haowen Wang, Mingsheng Long, 2024.

---

> > ### Author Response · Authors · 2025-11-22
> > **Answer to Reviewer HKTM (2)**
> >
> > ## W4: Conditioning on PDE parameters
> >
> > Thanks for the suggestion. Indeed, conditioning PDE parameters using MLPs and injecting them through AdaLN has already been explored in prior works [1, 2, 3]. We have now added the UniSolver reference for proper attribution (lines 264–265).
> > To clarify, we do not present this mechanism as a novelty. It is only used in our experiments to evaluate ECHO in a setting where the model receives a single initial condition rather than multiple past frames. In such a case, the dynamics cannot be inferred from the input alone, and the PDE parameters are required to specify a well-posed problem.
> > (corresponding to results in appendix E.3).
> >
> > ## W5: Details on three-stage training
> >
> > Thanks for the suggestion, we have included the modifications in the updated text (appendix D).
> > We now provide more hyper-parameters details on the three stage training in Section (appendix D). In order to clarify, we give a high-level description of the 3 stages below and are ready to explain in more details if needed.
> >
> > In the first low-resolution training stage (stage 1), we learn the encoder-decoder directly to reconstruct the full trajectory sequence, while  intentionally reducing the number of spatial points to fit GPU memory. The model is warmed-up using a linear strategy until reaching a learning rate of 1e-3, and then we use a cosine schedule (final learning rate is 1e-7). In the high-resolution stage (stage 2), we decrease the number of frames to process (according to GPU capacity), but increase the number of spatial points. This is a refinement stage, so we only train it for a few number of epochs with a learning rate of 1e-4 that decreases also following a cosine schedule. For  the generative process (stage 3), we freeze the encoder-decoder and train the processor. For all stages, we use an AdamW optimizer. All these details have been added in appendix D.
> >
> >
> > ## W6 Ablation on the sampling process
> > Concerning the sampling process, we did not provide ablations, as it was not the  focus of the paper. In the experiments, we used the midpoint method to solve the ODE (stated in line 187-188) with 10 denoising steps. As requested, we provide two additional ablations (section E.6.3), to show the impact of the number of Flow matching steps and the chosen solver. The results are reproduced below. They show that adding more steps mildly improves the results.
> >
> > | Ablation | Configuration | Rel. MSE ↓ |
> > |----------|----------------|-------------|
> > |
> > | **Solver** | Euler     | 1.62e-1     |
> > |         | Midpoint     | 1.67e-1     |
> > |         | RK4          | 1.68e-1     |
> > |         | DOPRI5       | 1.71e-1     |
> > ||
> > | **# Steps** | 2 steps      | 1.72e-1     ||
> > |          | 5 steps      | 1.69e-1     |
> > |          | 10 steps     | 1.68e-1     |
> > |          | 25 steps     | 1.67e-1     |
> >
> > ## W7: Additional dataset
> > We thank the reviewer for this suggestion. However, the DrivAerNet++ dataset is a static dataset that does not include temporal evolution, which is the focus of our paper. Following the reviewer’s recommendations, in order to demonstrate the scaling potential of ECHO, we have conducted additional experiments on two 3D benchmarks from The Well reference dataset, which better aligns with the objectives of our study. We provide the results below:
> >
> > | Dataset | Baseline | 1st Frame | 1st 8 Frames | 1st 16 Frames |
> > | :--- | :--- | :---: | :---: | :---: |
> > | **MHD** | FNO | 1.69e-1 | nan | nan |
> > | | AVIT | 0.34036 | 0.56809 | 0.75077 |
> > | | BCAT | 0.329 | 0.543 | 0.670 |
> > | | ECHO | 0.270 | 0.382| 0.550 |
> > | **Turbulence** | FNO | 3.15e-2 | nan | nan |
> > | | AVIT | 0.126 | 0.622| 1.319 |
> > | | BCAT | 0.124 | 0.989 | 1.621 |
> > | | ECHO | 0.113 | 0.555 | 0.667 |
> >
> > The performance demonstrates the ability of ECHO to scale to 3d datasets. We have added this additional experiments in table 5, section 4.5.
> >
> > We would like to highlight that in addition to the added experiments cited before, we also added an evaluation of ECHO in the interpolation task on $4$ datasets (section E4) and additional long-range generation (section 4.4.1) with a spectral analysis (section E1).
> > We hope this address all your concerns. We have modified the paper accordingly with all your feedbacks. We believe this improve the quality of the paper and better reflect the novelty and soundness of our work.
> > We thank the reviewer again for their valuable feedback and are happy to engage further if needed.

---

### Official Review · Reviewer_W4aJ · 2025-10-30

**Soundness:** 2
**Presentation:** 3
**Contribution:** 2
**Rating:** 4
**Confidence:** 4

**Summary:**

The paper proposes a generative latent transformer for spatio-temporal PDEs that encodes irregular meshes into a hierarchical spatio-temporal latent grid and then uses flow-matching to generate trajectories (rather than standard temporal autoregressive rollouts). This design makes global attention tractable at million-point scales, reduces long-horizon drift, and supports conditional tasks (forward prediction, interpolation, inverse/temporal conditioning). Experiments on several PDE benchmarks show the competitive performance of the model.

**Strengths:**

1. The paper is clearly written, which makes the method and implementation easy to follow.

2. Consistently lower MSE compared against other models across diverse PDE benchmarks that contains relatively large-scale problem.

3. The paper indicates that hierarchical convolutional yields lower reconstruction error at a fixed latent size than non-conv alternatives (graph, INR, and transformer-AE).

**Weaknesses:**

1. While the system is well-executed, many components are established (conv encoder/decoder, regular latent grid, DiT-style transformer, flow matching/latent diffusion). Related efforts already explore latent generative PDE solvers—e.g., [1] conv AE + structured latent grid with flow-matching DiT, [2] latent diffusion generating full trajectories at once, [3] autoregressive latent video diffusion.

A clearer exposition of the main differences would strengthen the paper’s contribution

[1] Li, Zijie, Anthony Zhou, and Amir Barati Farimani. "Generative Latent Neural PDE Solver using Flow Matching."

 [2] Zhou, Anthony, et al. "Text2pde: Latent diffusion models for accessible physics simulation."

 [3] Nguyen, Tung, et al. "PhysiX: A Foundation Model for Physics Simulations."

2. Generating the entire trajectory at once works well on cases like cylinder flow is not surprising, but it’s unclear how robust this design is for more turbulent regimes and beyond the training horizon. The appendix’s shallow-water example is reassuring; it would be helpful to see a similar long-horizon generalization test on a more turbulence case like the 2D vorticity problem in the benchmark to confirm the effect. holds.

**Questions:**

Lower MSE can reflect over-smoothing/blur, especially in turbulent fields. Could the authors report energy spectra  alongside MSE to verify how well the small-scale content is preserved?

---

> ### Author Response · Authors · 2025-11-22
> **Answer to Reviewer W4aJ (1)**
>
> We appreciate the reviewer’s positive feedback on the clarity of our manuscript as well as the experiments. We address the raised concerns below, and as suggested by the reviewer, we also provide additional experiments on long term horizon generalization for turbulent flow (2D vorticity) and we report and analyze the energy spectra of the forecasts.
>
> # Weaknesses
>
> ## W1: Innovative aspects of the contribution w.r.t. references
> Our work makes multiple distinct contributions that we restate in the following. The core motivation behind ECHO is to introduce a generative operator capable of one-shot trajectory synthesis for states containing millions of points. Crucially, ECHO achieves this while maintaining strict efficiency across three axes: accuracy, compression ratio, and inference latency. This regime reflects real-world constraints where current neural operators demonstrably fail (see Table 3 and 4). Unlike the cited references, ECHO is explicitly designed to resolve this specific efficiency-scalability bottleneck.
> ECHO addresses this via the following technical contributions:
>
> - **Efficient irregular to regular encoding**:
>     - [1,2] main innovation is respectively on designing generative processes for PDE solvers and on conditioning neural solvers (e.g. with text prompts) while adapting to arbitrary discretization. Although efficiency is present in these work, this is not their primary focus and the proposed methods do not scale to large size million point geometries problems as illustrated in table 4.
>     In contrast, to achieve our objective, we show that mesh points can be mapped onto a fixed regular latent grid with a continuous convolution operation, saving a lot of memory during training. For very dense latent grids, we further demonstrate that our encoding can be computed regions by regions, providing a practical solution to memory constraints at a low cost.
>     - [1, 2]  performs compression spatially. In contrast, we perform spatio-temporal compression, this is fundamental in our model since one generates trajectories in one shot and compressing the temporal dimension implies fewer tokens and yields significantly improved transformer scaling.
>
> - **Multi-task formulation for trajectory generation.**
> In [1], generation is auto-regressive, and thus suffers from the well-known issue of error accumulation over long rollouts.
> In [2], full-trajectory generation is supported, but conditioning is restricted to text prompts or initial conditions via AdaLN. In contrast, ECHO conditions directly on any arbitrary frame in the sequence. This enables the model to solve multiple downstream tasks—forward prediction, inverse problems, and interpolation—within a single unified generative framework. This design also allows ECHO to capture process dynamics without requiring explicit PDE models or parameters, enabling natural handling of parametric PDEs, unlike [2].
>
> - **Novel training strategy enabling million-point encoding** Our hierarchical multi-stage training procedure is specifically engineered to make million-point spatio-temporal encoding feasible. It ensures high-resolution fidelity under tight memory constraints, a regime unexplored in prior work.
>
> We thank the reviewer for pointing us to [3]. That work focuses on multi-physics forecasting and adopts an autoregressive next-token prediction objective, using a spatio-temporal tokenizer that quantizes physical inputs into discrete representations. In contrast, our generative process operates in a continuous space (without any quantization), supports irregular geometries, and enables multi-task generation—capabilities that fall outside the scope of [3].
>
> ## W2: Additional experiments for long horizon trajectory generation.
> We agree on the importance of trajectory extrapolation when generating multi-steps in one shot. As mentioned by the reviewer, we already provided an experiment showing the extrapolation capabilities of ECHO on Shallow-Water (Appendix E1). Also, note that Figure 1, center plot shows the error per time-step on Gray-Scott on 160 steps, while it has only been trained to predict 40 frames during training. The figure demonstrates that ECHO is extremely competitive compared to all other baselines.
>
> As suggested by the reviewer, we conducted an additional experiment on the 2D vorticity dataset. The model is trained on an horizon of 20 timesteps, and is evaluated on a horizon of $50$ timesteps. We report the results in the table below which shows that the performance degrades gracefully with the extrapolation horizon.
>
> |Horizon |0.5T | T|1.5T|2T|2.5T|
> |:-:|:-:|:-:|:-:|:-:|:-:|
> |Error|7.712e-2| 1.718e-1 | 2.618e-1 | 3.790e-1 | 4.7902e-1 |
>
> We added these results in the main text (figure 4 - section 4.4).

---

> ### Author Response · Authors · 2025-11-22
> **Answer to Reviewer W4aJ (2)**
>
> # Questions
>
> - **Q1: Additional experiments with energy spectra.**
>
> Thanks for the suggestion. We conducted a new experiment showing the energy spectra of the model predictions on the vorticity equation for Forward task. We report two plots of the energy spectra, respectively for predictions inside the training horizon and outside the training horizon. They have been inserted in the updated text in Appendix E.1.2.
> In both settings, ECHO reproduces the spectral slope and large- to mid-scale energy distribution with high fidelity. Although a slight excess of energy appears at very high wavenumbers—reflecting mild fine-scale discrepancies—this deviation remains bounded even in long rollouts. Overall, the model preserves the multiscale structure of the flow and demonstrates strong long-term stability: spectral accuracy remains high well beyond the training horizon, with very little drift or collapse of physical behavior.
>
> We would like to highlight that in addition to the added experiment cited above, we also added additional results on $3$D datasets to further evaluate the capacity of ECHO to operate on dense grids (section 4.5) , we evaluate echo on the interpolation tasks on 4 datasets (section E4) and we provide additional ablation on the generative components of ECHO (section E.6.3).
>
> We hope this address all your concerns. We have modified the paper accordingly with all your feedbacks. We believe this improve the quality of the paper and better reflect the novelty and soundness of our work.
> We thank the reviewer again for their valuable feedback and are happy to engage discussion further if needed.
>
> [1] Li, Zijie, Anthony Zhou, and Amir Barati Farimani. "Generative Latent Neural PDE Solver using Flow Matching."
>
> [2] Zhou, Anthony, et al. "Text2pde: Latent diffusion models for accessible physics simulation.
>
> [3] Nguyen, Tung, et al. "PhysiX: A Foundation Model for Physics Simulations."

---

### Official Review · Reviewer_QwMS · 2025-11-01

**Soundness:** 2
**Presentation:** 2
**Contribution:** 2
**Rating:** 4
**Confidence:** 4

**Summary:**

The model adopts an encoder-decoder architecture, enabling efficient solving of large-scale partial differential equations with millions of points. A specialized encoder is designed to map irregular grids in physical space to regular grids in latent space, thereby simplifying the downsampling process for large-scale point sets. At its core, the model employs a DiT-based generative approach. After training, it possesses multi-task capabilities, allowing it to perform forward and inverse solving along the temporal dimension as well as data interpolation tasks simultaneously.

**Strengths:**

1. It employs a compression-decompression architecture, supporting the solution of PDEs with millions of points.
2. The designed encoder can map irregular grids in physical space to regular grids in latent space.
3. Using a DiT-based generative approach, the trained model can simultaneously support multiple tasks such as forward solving, inverse solving, and interpolation.
4. By generating complete trajectory segments, it effectively mitigates long-term error accumulation.

**Weaknesses:**

1. One of the contributions of the paper is its support for predictions with millions of spatial points. However, the compression process of spatial points in the encoder and decoder is not described in detail. Appendix C.2 mentions that the authors adopted structures similar to (Hagnberger et al., 2025), (Yu et al., 2023), and (Koupaï et al., 2025). So, what are the innovative aspects of the authors' work in spatiotemporal compression and decompression that distinguish it from these references?
2. In Table 1, the spatial dimensions of all datasets are 2D. Neural operators supporting millions of spatial points are highly suitable for tasks involving large-scale 3D point clouds, and most real-world PDE problems are 3D. Therefore, exploring the applicability of this operator to irregular 3D point clouds would be highly meaningful.
3. The algorithm can handle multiple tasks such as forward solving, inverse solving, and interpolation. However, the interpolation task lacks dedicated experimental support in Table 2 or Table 4.
4. In Section 2.1 and Appendix C.1, the forward, inverse, and interpolation tasks are defined along the temporal dimension. In Table 4, when X_te=20%, the model performs an interpolation task in the spatial dimension. How spatial interpolation is achieved lacks a clear explanation.
5. Descriptions of certain tasks are insufficient. In Table 4, what do Time-Stepping (TS) and Reconstruction (Rec) refer to, and how are they specifically executed?
6. The Related Works section should be included in the main body of the paper rather than in the appendix.

**Questions:**

1.The description of the Encoder and Decoder methods lacks clarity. The formulas and figures should be supplemented to better illustrate the process. The dimensionalities of variables and how they change throughout the process are not clear. The authors should provide a more detailed explanation in the main text or the appendix.
2.In lines 289-299, the spatial downsampling rate during training is stated to be between 0.5 and 1.0. However, in Table 4, the spatial sampling ratio is 20%, which is less than 50%. The reason for this inconsistency in the ranges should be clarified.
3.In Line 160, "ODE" likely refers to "Ordinary Differential Equation." The full term should be written out upon its first appearance.
4.For the "Acoustic Scattering Maze" dataset in Table 1, the corresponding reference citation is missing and should be added.
5.The references for the AROMAF and CALM-PDE methods are currently cited as arXiv preprints. Since they have now been formally published, the citations should be updated to their final, officially published versions. The citations for the other methods should also be checked for accuracy and completeness.

---

> ### Author Response · Authors · 2025-11-22
> **Answer to Reviewer QwMS (1)**
>
> We would like to thank reviewer QwMS for its detailed review and suggestions to improve the clarity of our paper. We address the raised concerns below. We have also added the results of new experiments on 3D simulations performed on 2 datasets and an interpolation evaluation as suggested by the reviewer.
>
> # Weaknesses
> ## W1: Innovative aspects of the contribution w.r.t. references (Koupaï et al., Hagnberger et al.).
> We acknowledge that our prior phrasing was ambiguous. By 'similar,' we intended to highlight that these models share a high-level encode-process-decode framework and utilize hierarchical convolutional compression. However, both our core objectives and technical contributions differ significantly, as we have now clarified in the revised manuscript.
>
> Regarding the objectives, ECHO specifically targets efficient support for million-point geometries. We define 'efficiency' here as the ability to compress dense physical inputs into highly compact latent representations without sacrificing accuracy or inference speed. As demonstrated in Table 4 and Figure 1, the cited methods [2, 3, 4] do not address this specific efficiency aspect, failing to scale or maintain physical fidelity at high compression ratios. ECHO overcomes these limitations through the following technical contributions:
>
> - **Efficient irregular to regular encoding.** In [1, 2, 3], mesh points are mapped to an iteratively learned latent grid. This approach necessitates storing the entire computational graph at each iteration, inducing substantial memory and computational overhead that becomes prohibitive for large or dense grids. In contrast, ECHO circumvents this bottleneck by mapping mesh points onto a fixed regular latent grid. This design significantly reduces training memory requirements and allows for region-wise (chunked) computation on very dense grids, effectively solving memory constraints with minimal overhead. Furthermore, the resulting regular grid structure enables the use of standard, highly efficient convolutional operators for compression.
>
> -  **Hierarchical compression**: Due to these memory constraints, methods [1, 2] are forced to apply aggressive compression ratios in their first compression block to fit within GPU memory, resulting in significant information loss (see ablation study in Table 20, Appendix E.6.1). In contrast, ECHO’s efficient latent mapping enables a deep hierarchical compression strategy. This approach preserves relevant physical information even at extreme compression ratios (up to ×256), as demonstrated in Figure 1 (left).
>
> - **Spatio-temporal compression**: While methods [1, 3] leverage only spatial compression, ECHO implements joint spatio-temporal compression. This distinction is fundamental to our design: by compressing the temporal dimension, we significantly reduce the total token count processed by the transformer. This reduction is precisely what enables our 'one-shot' generation paradigm, allowing the model to predict entire trajectories simultaneously with improved computational scaling.
>
> - **Mesh independence**: Regarding the comparison with [4], we note that this method is restricted to regular grids and rely on quantization due to its focus on discrete video generation. In the context of spatiotemporal physical dynamics, however, mesh independence is a highly desirable property. Unlike [4], ECHO supports arbitrary irregular domains in a continuous space, which enables us to perform complex spatial tasks (such as querying off-grid locations or super-resolution) that grid-locked video models cannot handle.
>
> These technical considerations allow ECHO to generate million point PDE trajectories in a very compact latent space and preserve fidelity (Table 3.) and low latency (Table 17).
>
> [1] Universal Physics Transformers: A Framework For Efficiently Scaling Neural Operators, Benedikt Alkin, Andreas Fürst, Simon Schmid, Lukas Gruber, Markus Holzleitner, Johannes Brandstetter, 2024.
>
> [2] Armand Kassaï Koupaï, Lise Le Boudec, Louis Serrano, and Patrick Gallinari. ENMA: Tokenwise autoregression for continuous neural PDE operators, 2025.
>
> [3] Jan Hagnberger, Daniel Musekamp, and Mathias Niepert. CALM-PDE: Continuous and adaptive convolutions for latent space modeling of time-dependent PDEs, 2025.
>
> [4] Lijun Yu, Yong Cheng, Kihyuk Sohn, José Lezama, Han Zhang, Huiwen Chang, Alexander GHauptmann, Ming-Hsuan Yang, Yuan Hao, Irfan Essa, et al. Magvit: Masked generative video transformer, 2023.

---

> > ### Author Response · Authors · 2025-11-22
> > **Answer to Reviewer QwMS (2)**
> >
> > ## W2: Additional experiments for 3D evaluation
> >
> > We appreciate the suggestion. To the best of our knowledge, there are currently no established benchmarks for 3D irregular spatio-temporal dynamics in the literature. Consequently, to demonstrate ECHO's scalability to 3D environments, we incorporated **two challenging 3D datasets** from 'The Well': Magnetohydrodynamics (MHD) and Turbulence Gravity Cooling (TGC). In these experiments, the model predicts 16 subsequent frames based on a 4-frame context. The table below reports the relative MSE averaged over the 1st, 8th, and 16th predicted frames.
> >
> > | Dataset | Baseline | 1st Frame | 1st 8 Frames | 1st 16 Frames |
> > | :--- | :--- | :---: | :---: | :---: |
> > | **MHD** | FNO | 1.69e-1 | nan | nan |
> > | | AVIT | 0.34036 | 0.56809 | 0.75077 |
> > | | BCAT | 0.329 | 0.543 | 0.670 |
> > | | ECHO | 0.270 | 0.382 | 0.550 |
> > | **Turbulence** | FNO | 3.15e-2 | nan | nan |
> > | | AVIT | 0.126 | 0.622 | 1.319 |
> > | | BCAT | 0.124 | 0.989 | 1.621 |
> > | | ECHO | 0.113 | 0.555 | 0.667 |
> >
> > This demonstrates that ECHO achieves accurate forecasts also on the 3D cases, and outperforms the baselines most of the time, showing its potential for large scale 3D+1 spatio-temporal data, with a compression ratio of $\times 112$, while the baselines perform in the physical space at a higher computational cost. These results have been added to the updated manuscript in table 5 - section 4.5.
> >
> > ## W3:  Additional experiments for the interpolation task
> >
> > Following your suggestion, we evaluated ECHO on temporal interpolation by reconstructing intermediate frames (11 frames for standard datasets, 16 for Active Matter, and 36 for Gray-Scott). Since standard baselines are autoregressive and cannot perform direct interpolation without significant modification, we benchmarked against a deterministic variant of ECHO.
> > Furthermore, since most competing methods (excluding ENMA) are deterministic, this comparison effectively highlights the advantage of our generative formulation. The results are presented below and have been added in the manuscript in Table 17 (Appendix E.4) column "Interp.". This illustrates ECHO's effectiveness in this setting.
> >
> > |Baseline | AM | GS | ASM | RB |
> > |-----|-----|-----|-----|-----|
> > |ECHO deterministic| 1.47e-1 | 3.12e-2 | 1.88e-1 | 1.08e-1|
> > |ECHO generative|  1.11e-1 | 1.781e-2 | 1.00e-1 | 9.02e-2|
> >
> > ## W4: Description of spatial interpolation
> > We thank the reviewer for this comment that gives us the opportunity to clarify this point. Spatial interpolation is intrinsic to our decoder design, which mirrors the encoder's hierarchical structure (lines 245-248). Specifically, the decoder utilizes a continuous convolution layer as its final stage. This mechanism enables the model to map latent representations back to the physical space at arbitrary query locations, thereby performing interpolation naturally. We have clarified this description in the revised manuscript (lines 245-248 and Appendix C.1).
> >
> > ## W5: Descriptions of tasks
> >
> > Thanks for pointing out this. This is addressed in the updated version of the paper ('setting' paragraphs of sections 4.2,4.3 and appendix C) and we provide a synthetic description of the tasks below.
> >
> > The experiments in Table 3 (formerly table 4) are designed to assess the advantage of our processor design, which generates entire trajectory segments in one shot, over state-of-the-art baselines. We benchmarked performance on both forward tasks (conditioning on the initial 4 frames) and backward tasks (conditioning on the final 4 frames). The results, reported as Relative L2 and latency in table 2, demonstrate the robustness of our Transformer design in handling multi-tasks efficiently.
> >
> > In Table 3 (formerly Table 4), our objective is to benchmark state-of-the-art architectures designed to map irregular meshes into compact latent spaces. We evaluate the capacity of each encoder-decoder pair using two distinct metrics:
> >
> > - **Reconstruction Quality ('Rec')**: We first trained each encoder-decoder on a pure reconstruction objective and measured fidelity using Relative MSE. To ensure a fair comparison, we fixed the latent space dimensionality across all methods, noting exceptions where Out-Of-Memory (OOM) constraints necessitated size reductions (indicated with colors).
> >
> > - **Latent Space Expressivity ('For')**: We assessed the suitability of the learned latent spaces for dynamics modeling via a forward forecasting task. To isolate the quality of the encoding, we utilized the ECHO processor across all experiments. The processor was trained individually on the latent representations produced by each baseline, using 4 context frames to predict the remainder of the trajectory.
> >
> > ## W6: Related work section
> >
> > We acknowledge the point on related work placement, but we had to make a compromise given the limited amount of space. We present a synthetic description of the state of art in the introduction while providing more details in the appendix A.

---

> > > ### Author Response · Authors · 2025-11-22
> > > **Answer to Reviewer QwMS (3)**
> > >
> > > # Questions
> > >
> > > - **Q1**: Thanks for the remark. We have updated the description of Fig 2 and 3. We provide more details about the model in the appendix. We have also added lines 196-198 a description of the pipeline with an indication of the shape values and we have updated Figure 2. We also provide more details on the model in Appendix C. We hope that this clarifies the model description.
> > >
> > > - **Q2**: Thanks for the question. We probably were not clear enough and will take care to better formulate in the manuscript (lines 419-426).
> > > During training, we intentionally reduce the number of spatial points; this is a technical choice to reduce memory compute and help the model to better generalize to scenarios with few number of spatial points at inference. We set it between 20 to 50 percent at stage 1 and 50 to 100 percent at stage 2.
> > > In Table 4, we then evaluate the model with 100\% and 20\% of the inputs points, which is aligned with the training setup. In Table 15, we also test it on the 10\% scenario and show that ECHO performs competitively or better than existing baselines on this setting not encountered at training time.
> > > As a complement, note that the proposed model can be queried at any point. Said otherwise, whatever the input sample, the model can be queried on the whole spatial domain of the problem. This is a desirable property of the continuous convolution used in the decoder (see also W4). Then the model can be trained with a given ratio of input points and be evaluated at other ratios.
> > >
> > > - **Q3**: Thanks for pointing that out. We added the ”ODE” reference in the revised version (line 160).
> > >
> > > - **Q4**: Thanks, this has been be corrected. This is a dataset from The Well. It was a typo error (see table 1).
> > >
> > > - **Q5**: Thanks for pointing this out. CALM-PDE has been accepted very recently at NeurIPS and we have checked all citations and made sure that they are up-to-date.
> > >
> > > We would like to highlight that in addition to the added experiment decribed above, we also added long-range rollout on the Vorticity dataset and a spectral analysis on this generation as well as new ablations on the generative components of ECHO.
> > > We hope this address all your concerns. We have modified the paper accordingly with all your feedbacks. We believe this improve the quality of the paper and better reflect the novelty and soundness of our work.
> > > We thank the reviewer again for their valuable feedback and are happy to engage further if needed.

---

### Author Response · Authors · 2025-11-22
**General Answer to Reviewers - additional experiments and modifications**

We thank all reviewers for the time spent reading and evaluating our manuscript. Their comments and suggestions have greatly helped us improve the paper. Below, we summarize the main modifications. Following the reviewers’ requests, we have also conducted several additional experiments—including evaluations on two $3$D datasets, temporal interpolation, long-horizon turbulent forecasting, spectral analyses, and ablations — which have now been incorporated either in the main text or the appendix. For clarity, they are highlighted in red in the new version of the paper.

- We now propose an **additional experiment on 2 3D dataset** (Reviewers QwMS and HKTM - section 4.5 of the updated manuscript)
- We conducted a **complete comparison on the temporal interpolation task** (Reviewers QwMS and uXAC appendix E.4)
- We evaluated the **long-range capabilities in a turbulent case** (ie on the Vorticity dataset, up to $50$ timesteps) in section 4.4 and we provide a **spectral analysis of the generated trajectories** (appendix E1) (reviewer W4aJ)
- We added some **ablation studies on the sampling components of ECHO** (number of steps, solver) (appendix E.6.3) (reviewer HKTM)
- We have **clarified the encoding shapes** in the main text (lines 195-197 and figure 2) (reviewer QwMS)
- We have more **clearly detailed the experimental setting** in the main text (section 4 and appendix D) (Reviewers QwMS, HKTM and uXAC)
- We have **added several implementation details** in the  in the appendices (appendix C and D) (Reviewers QwMS, HKTM and uXAC)
- We provide **additional visualization of generated trajectories** with ECHO (appendix F)

We hope our responses and additional experiments address the reviewers concerns. We would be happy to further engage in discussion if needed.

---

### Author Response · Authors · 2025-12-04
**Final Answer to AC - Summary of the work and Discussion Period (1/3)**

Dear AC,

With the additional responsibilities assigned to ACs —including reassessing how reviewer feedback might have evolved after the rebuttal—we provide below a concise summary of our paper and of our responses to the reviewers’ comments and questions to help streamline your assessment. We also want to emphasize that, beyond technical clarifications, we have added a substantial set of complementary experiments addressing each reviewer suggestion.

## Summary of the paper

We introduce ECHO, a Transformer-based neural operator that generates full trajectories in a single shot on very dense meshes (up to one million spatial points per frame). The paper is motivated by an experimental analysis in Fig. 1, where we identify several failure modes of existing autoregressive neural surrogates operating in compressed latent spaces:
F1) They cannot preserve physical information under strong compression ratios.
F2) They suffer from pronounced error accumulation.
F3) They underperform on diverse temporal tasks.

We provide intuitions for these limitations of current models and propose ECHO to overcome these challenges through three main technical contributions:

### 1) Hierarchical spatio-temporal encoder–decoder (F1).
ECHO iteratively compresses dense meshes into a compact latent using a convolution-only architecture that preserves physical structure, retaining more information than prior neural operators across compression ratios (Fig. 1 left, Table 3). Ablation §3.6.1 shows that fidelity is strongly influenced by the number of hierarchical blocks, and ECHO uniquely supports full-resolution queries on very dense trajectories (Table 4).

### 2) Refinement through a two stage training process  (F1).
A low-resolution trajectory training phase followed by high-resolution frame-wise refinement enables efficient training on dense meshes. Ablation §E.6.2 quantifies the benefit of this refinement stage.

### 3) Generative one-shot trajectory model (F2 & F3).
ECHO outputs full trajectories in one shot, conditioned on an arbitrary number of frames, enabling forward, inverse, and interpolation tasks within a single model. In Table 2, it is shown to outperform both physical- and latent-space neural solvers, generative and deterministic, on forward and inverse tasks.


Across the four reviews, the paper was initially viewed as interesting and potentially impactful.
Reviewers asked for more details regarding the positioning and explanation of our novelty relative to prior latent generative PDE models, additional experiments (interpolation, 3D, long-horizon turbulence), and more technical detail. These comments led to a revision of the paper, including substantial new experiments that directly address each reviewer suggestion, as well as methodological and technical clarifications. We believe the reviewers’ feedback has considerably strengthened the submission.


Below, we group the strengths and weaknesses raised by all reviewers by theme. For each weakness, we summarize our response and the additional experiments we performed to address it.

## Strengths
### Methodological novelty & design
- Efficient hierarchical spatio-temporal compression for irregular grids. (QwMS, uXAC)
- Flow-matching–based generative formulation enabling forward, inverse, and interpolation tasks within a single model. (QwMS, uXAC)
- Convolutional hierarchical design that yields better reconstructions than non-convolutional alternatives. (W4aJ, uXAC)
### Experiments
- Extensive experiments with strong performance across diverse PDE systems, including large-scale and complex geometries. (W4aJ, HKTM, uXAC)
- Ability to operate at million-point resolution where most baselines fail. (QwMS, HKTM, uXAC)
- Solid experimental comparisons with state-of-the-art methods, with consistently lower MSE and reduced error accumulation. (W4aJ, HKTM)
### Presentation & scope
- Manuscript viewed as clear, easy to follow, and straightforward to implement. (W4aJ, uXAC)
- Well-motivated training strategy. (HKTM)
- Broad applicability and potential as a foundation model for PDE solving. (HKTM)

---

> ### Author Response · Authors · 2025-12-04
> **Final Answer to AC - Summary of the work and Discussion Period (2/3)**
>
> ## Weaknesses
>
> ### Novelty
> - Insufficient explanation of what is novel in hierarchical compression vs prior works.  (QwMS, W4aJ).
>
> **Response**. We clarified how ECHO differs from prior hierarchical approaches:
> 1) It encodes irregular meshes into a fixed regular grid, enabling large-scale physical systems to be processed region by region without exceeding memory.
> 2) It performs joint spatio-temporal compression, which is critical for one-shot trajectory generation, when others only focus on spatial compression.
> 3) It supports extremely high compression ratios (up to ×256 in space–time) while maintaining fidelity.
>
> - Relation to existing latent generative PDE solvers (W4aJ)
>
> **Response**.
> 1) Prior latent generative PDE models generally rely on autoregressive rollout, use less efficient representations of the physical signal, or adopt restrictive conditioning mechanisms that prevent them from scaling to million-point grids. In contrast, ECHO combines efficient spatio-temporal compression with a generative formulation operating directly on continuous latent fields. This enables one-shot generation of full trajectories while maintaining scalability on dense meshes.
> 2)  Through its specific instantiation of the generative model, ECHO supports multi-tasking when other latent models are most often limited to one task, typically forecasting..
>
> - Clarifications on continuous CNN vs GNN (HKTM)
>
> **Response.**  We expanded the discussion on the benefits of using continuous convolutions over GNNs: GNNs require explicit adjacency structures and become memory-heavy at the million-point scale. They tend to overfit to the mesh topology and cannot easily query arbitrary points. Continuous convolutions avoid these limitations and naturally operate on continuous spatial domains, which is better aligned with PDE settings.
>
> - Uniform latent grids & spherical domains (HKTM)
>
> **Response.** We clarified how ECHO handles spherical and complex domains: ECHO adopts coordinate transformations (e.g., spherical coordinates), and is not confronted to empty regions and their potential additional computational load. Empirical results on spherical (shallow water) and complex domains (eagle and cylinder-flow) confirm the effectiveness of uniform latent grids under these transformations.
>
> -  Conditioning of the generative model & PDE parameter handling (HKTM)
>
> **Response.** In the revised manuscript, we explain how the conditioning with masked tokens, allows us to handle our multitask setting and to our knowledge this is the first time it is used in this PDE setting. The model also allows the conditioning with PDE parameters. This is used only in specific cases, and is not positioned as an original contribution.
>
> ### Experiments
>
> In all tables, metric is the Relative L2 (lower is better).
>
> - Lack of 3D experiments (QwMS, HKTM)
>
> **Response.** We added new results on two large 3D datasets from the Well (MHD and TGC), showing that ECHO scales to large, challenging 3D systems. ECHO outperforms FNO, AVIT, and BCAT on most metrics, confirming its scalability. These results complement the original large-scale experiments at 1024×1024 resolution, which HKTM already described as “impressive”. We provide below the corresponding experimental evaluation, positioned in §4.5.
>
> | Dataset | Baseline | prediction horizon 1 Frame | prediction horizon 8 Frames | prediction horizon 16 Frames |
> | :--- | :--- | :---: | :---: | :---: |
> | **MHD** | FNO | 1.69e-1 | nan | nan |
> | | AVIT | 0.34036 | 0.56809 | 0.75077 |
> | | BCAT | 0.329 | 0.543 | 0.670 |
> | | ECHO | 0.270 | 0.382 | 0.550 |
> | **Turbulence (TGC)** | FNO | 3.15e-2 | nan | nan |
> | | AVIT | 0.126 | 0.622 | 1.319 |
> | | BCAT | 0.124 | 0.989 | 1.621 |
> | | ECHO | 0.113 | 0.555 | 0.667 |

---

> ### Author Response · Authors · 2025-12-04
> **Final Answer to AC - Summary of the work and Discussion Period (3/3)**
>
> - Lack of interpolation experiments (QwMS, uXAC)
>
> **Response.** We acknowledged that interpolation was under-emphasized and added new dedicated interpolation experiments (appendix §E4), including a controlled comparison between generative ECHO and a deterministic variant on 4 datasets from The WELL.
>
> |Baseline | Active Matter| Gray Scott | Acoustic Scattering Maze| Rayleigh-Bénard |
> |-----|-----|-----|-----|-----|
> |ECHO deterministic| 1.47e-1 | 3.12e-2 | 1.88e-1 | 1.08e-1|
> |ECHO generative|  1.11e-1 | 1.781e-2 | 1.00e-1 | 9.02e-2|
>
> - No long-range generation in turbulent regimes (W4aJ)
>
> **Response.** In addition to the original long-range OOD experiments, we added a new long-range generation experiment on the turbulent Vorticity equation (Section 4.4.1) demonstrating that the performance degrades gracefully with the horizon size (0.5T, T, 1.5T, 2T, 2.5T, with T the size of the prediction horizon seen during training). Additionally, we analyzed the energy spectra of the predictions and we provide the resulting figure in Appendix §E.1.2. ECHO faithfully reconstructs the physics of the system across the most energy-dominant scales. The only discrepancy is a slight overestimation of energy at the very finest scales (high spatial frequencies), where most neural surrogates are known to fail.
>
> |Horizon |0.5T | T|1.5T|2T|2.5T|
> |:-:|:-:|:-:|:-:|:-:|:-:|
> |Error|7.712e-2| 1.718e-1 | 2.618e-1 | 3.790e-1 | 4.7902e-1 |
>
> - Ablations on the sampling process (HKTM)
>
> **Response.** We introduced new ablations of the Flow-Matching sampling strategy (appendix §E63), varying both the number of denoising steps and the ODE solver.
>  Ablation | Configuration | Rel. MSE ↓ |
> |----------|----------------|-------------|
> |
> | **Solver** | Euler     | 1.62e-1     |
> |         | Midpoint     | 1.67e-1     |
> |         | RK4          | 1.68e-1     |
> |         | DOPRI5       | 1.71e-1     |
> ||
> | **# Steps** | 2 steps      | 1.72e-1     ||
> |          | 5 steps      | 1.69e-1     |
> |          | 10 steps     | 1.68e-1     |
> |          | 25 steps     | 1.67e-1     |
>
> These ablations show that ECHO is robust to the solver choice and benefits moderately from more denoising steps, without requiring very long sampling schedules.
>
>
> - Lack of metrics to evaluate efficiency (uXAC)
>
> **Response.** This concern is already addressed in the initial manuscript. In Table 2, we evaluate latency (time to generate a full trajectory) and show that ECHO is competitive with deterministic AR baselines on regular grids. In Appendix E.2, we report training throughput and inference latency on irregular grids, again showing that ECHO is competitive while still preserving physical information under different compression ratios (Fig. 1 left).
>
> - Evaluation with Transolver++ (uXAC)
>
> **Response.** We conducted an additional experiment with Transolver++ that shows that the baseline cannot generate a full trajectory with dense meshes trajectories (up to 20M points - which is the considered experiment). This result is now added in Table 4. The latter experiment specifically targets baselines that operate within a compact latent space and support resolution-independent querying. As Transolver++ processes data directly in the physical space and lacks the architectural flexibility to be queried at arbitrary output resolutions, it was initially excluded from this specific comparison.
>
> | Task   | GINO | CORAL | AROMA | CALM-PDE | Transolver++ | ECHO      |
> |:------|:----:|:-----:|:-----:|:--------:|:------------:|:---------:|
> | *For.* | OOM  |  OOM  |  OOM  |   OOM    |     OOM      | **3.88e-1** |
>
> - Evaluation on DrivAerNet++ (HKTM)
>
> **Response.** DrivAerNet++ is a static dataset without temporal dynamics, so it does not match ECHO’s focus on trajectory generation. Instead, we chose to add large-scale 3D temporal datasets, which better reflect the intended use case of ECHO.
>
> ## Presentation
> - Task descriptions and settings (QwMS, uXAC)
> **Response.** We clarified all experimental settings in the main text. We rewrote Appendix C for clearer task descriptions. We added more implementation details in Appendix D.
>
> - Typos and clarity (xUAC)
>
> **Response.** We corrected all reported typos. We improved wording and clarity in several sections throughout the manuscript. We also checked that all references are up-to-date.
>
> - Vorticity in the “irregular” section (uXAC)
>
> **Response.** We clarified why the Vorticity setup appears in the irregular-grid section.
>
> - Training strategy clarification (R1/QwMS, R3/HKTM, R4/uXAC)
>
> **Response.** The three-stage training process is now fully documented in Appendix D.
>
> We again thank the AC for the time and effort devoted to reassessing our work in this unusual context. The rebuttal process led to many new experiments, analyses, and revisions, and we believe we have now fully addressed all concerns raised by the reviewers.

---

### Meta-Review · Area_Chair_REGy · 2025-12-29

**Summary:**

I find this paper makes a good contribution to neural PDE surrogates at scale. ECHO combines hierarchical spatio-temporal compression with flow-matching generative modeling to achieve million-point trajectory generation where existing methods fail due to memory constraints. The core technical insight—that jointly compressing space and time enables tractable one-shot generation while reducing error accumulation—is convincingly demonstrated across diverse benchmarks. The rebuttal strengthened the submission substantially, adding 3D experiments on MHD and turbulence datasets, spectral analysis confirming physical fidelity, and ablations on the generative process.

I am not fully convinced the methodological novelty reaches the level of a top venue. The individual components (continuous convolutions, hierarchical CNNs, DiT-style flow-matching) are established, and the contribution is primarily in integration and engineering. However, the practical demonstration of scalability to millions of points, the consistent empirical improvements, and the responsive addition of requested experiments should be commended. The writing quality issues noted by reviewers suggest the submission was rushed, and I believe that a revised version of the paper with stronger methodological contributions will be a competitive future submission.

**Reviewer Concerns:**

Reviewer QwMS raised concerns about novelty relative to prior hierarchical approaches and requested 3D experiments. The rebuttal clarified how ECHO differs from prior work—specifically, joint spatio-temporal compression and the ability to achieve extreme compression ratios (up to 256×) while maintaining fidelity. The new 3D results on MHD and TGC datasets demonstrate scalability, though this evaluation remains limited to two datasets.

Reviewer W4aJ questioned robustness in turbulent regimes and requested spectral analysis. The authors provided vorticity extrapolation experiments (out to 2.5× the training horizon) and energy spectra plots. These show ECHO faithfully reconstructs low-to-mid frequency content, with only slight overestimation at the finest scales—consistent with known limitations of neural surrogates. I think this concern is adequately addressed.

Reviewer HKTM requested ablations on the sampling process and clarification on continuous convolutions versus GNNs. The rebuttal added Table 21 showing robustness to solver choice and step count, and provided a reasonable explanation of why continuous convolutions avoid mesh bias and scale better than GNNs. The DrivAerNet++ suggestion was appropriately declined as that dataset lacks temporal dynamics.

Reviewer uXAC gave the lowest score (2) citing missing interpolation experiments, unclear experimental design, and typos. Many of these concerns were already partially addressed in the original paper or have now been fully addressed. The Transolver++ comparison showed OOM on the million-point task, validating the scalability claims. I discount this rating somewhat as the criticism appears harsher than warranted given the acknowledged strengths.

**Reviewer Scores:**

Reviewer QwMS: 4 → 5. Major concerns (3D experiments, interpolation, task descriptions) were addressed with new experiments and clarifications. Should be satisfied.

Reviewer W4aJ: 4 → 5. Spectral analysis and long-horizon turbulent experiments directly answer the key technical questions raised. Likely to move toward acceptance.

Reviewer HKTM: 4 → 5. Ablations on sampling process added, training details expanded in Appendix D. Residual concerns are minor.

Reviewer uXAC: 2 → 4. The criticisms were either already addressed or are now explicitly addressed in the rebuttal. The original score may be too harsh given the strengths acknowledged.

---

### Decision · Program_Chairs · 2026-01-26

Reject